# How Far Can Camels Go? Exploring the State of Instruction Tuning on Open Resources

**Yizhong Wang**[*♣♠]    **Hamish Ivison**[*♣]    **Pradeep Dasigi**[♣]    **Jack Hessel**[♣]
**Tushar Khot**[♣]  **Khyathi Raghavi Chandu**[♣]  **David Wadden**[♣]  **Kelsey MacMillan**[♣]
**Noah A. Smith**[♣♠]    **Iz Beltagy**[♣]    **Hannaneh Hajishirzi**[♣♠]

[♣]Allen Institute for AI   [♠]University of Washington
{yizhongw,hamishi}@allenai.org

## Abstract

In this work we explore recent advances in instruction-tuning language models on a range of open instruction-following datasets. Despite recent claims that open models can be on par with state-of-the-art proprietary models, these claims are often accompanied by limited evaluation, making it difficult to compare models across the board and determine the utility of various resources. We provide a large set of instruction-tuned models from 6.7B to 65B parameters in size, trained on 12 instruction datasets ranging from manually curated (e.g., OpenAssistant) to synthetic and distilled (e.g., Alpaca) and systematically evaluate them on their factual knowledge, reasoning, multilinguality, coding, safety, and open-ended instruction following abilities through a collection of automatic, model-based, and human-based metrics. We further introduce TÜLU 🐪, our best performing instruction-tuned model suite finetuned on a combination of high-quality open resources.

Our experiments show that different instruction-tuning datasets can uncover or enhance specific skills, while no single dataset (or combination) provides the best performance across all evaluations. Interestingly, we find that model and human preference-based evaluations fail to reflect differences in model capabilities exposed by benchmark-based evaluations, suggesting the need for the type of systemic evaluation performed in this work. Our evaluations show that the best model in any given evaluation reaches on average 87% of ChatGPT performance, and 73% of GPT-4 performance, suggesting that further investment in building better base models and instruction-tuning data is required to close the gap. We release our instruction-tuned models, including a fully finetuned 65B TÜLU 🐪, along with our code, data, and evaluation framework to facilitate future research.[2]

## 1 Introduction

The latest generation of large language models has brought unprecedented attention to the potential of language technologies. To support imperative user requests and a chat interface, these models often undergo an *instruction-tuning* step which involves training on supervised input/output pairs. Recent instruction tuning corpora are often gathered via crowdsourcing (Dolly [12], Open Assistant [26]) or via distillation from another model (Alpaca [43], Vicuna [8]). However, while some public, instruction-tuned models are advertised as comparable to powerful closed-source proprietary models such as ChatGPT, most experiments that support such claims only cover a small set of tasks, and mostly rely on model-based evaluation metrics [8, 56]. We contend that the evaluation setup should

---

[*]Equal contribution.
[2]https://github.com/allenai/open-instruct

include tasks that test core reasoning and fact-recall skills of the model, in addition to testing model- or human-annotated generation quality, which may be more open-ended and subjective.

This paper provides a comprehensive evaluation of instruction-tuning resources: specifically, we conduct a large number of instruction tuning experiments spanning a dozen public corpora, and models ranging in scale from 6.7B to 65B. We evaluate both specific model capabilities (i.e., factual knowledge, reasoning, multilinguality, coding, safety) and open-ended instruction-following abilities. We report results based on automatic, model-based, and human-based evaluation metrics.

Our evaluation reveals that instruction tuning over different datasets appears to promote specific skills, and no one dataset provides the best performance across all evaluations. We also find that the underlying base model is paramount, with better base models (whether it be models trained on more tokens or larger models) performing better across the board. Surprisingly, we also find that the best-performing models in model-based evaluation are not the same as those that perform best on benchmark-based automatic evaluations, potentially partially due to GPT-4's strong bias toward long, diverse generations.

Building on our findings, we introduce TÜLU 🐪, a suite of 7B to 65B LLAMA models finetuned on a combination of data sources. TÜLU 🐪 65B is the largest publicly-released fully-instruction tuned LLAMA variant at the time of writing, to the best of the authors' knowledge. It is trained on 7 popular available datasets, and yields the best average performance across most model sizes while remaining within 29% of the best-performing model on each individual task. In summary, our key findings include:

- Instruction datasets targeted at specific domains and/or capabilities are extremely effective at improving model performance in those aspects.
- Larger or pretrained-for-longer base models consistently perform better than smaller ones after instruction tuning.
- Our model TÜLU 🐪 – fine-tuned LLaMa on a combination of existing instruction datasets – achieves the best average performance across benchmarks, although it is not the overall best when considering different evaluation settings independently.
- Even a very large (65B) model finetuned on a large mix of instruction datasets fails to outperform ChatGPT, although it does perform significantly better than similar smaller models.
- Model-based preference evaluation on open-ended instruction following correlates strongly with the average number of unique tokens generated by a model, suggesting that model-based preference evaluation has biases that may hide differences in model capabilities.

We open-source the code for training and evaluating these large language models. We also release checkpoints trained on the different instruction datasets and their mixtures, including TÜLU 🐪. We hope this facilitates further development and investigation of open instruction-tuned models.

## 2 Background: Instruction Tuning and Resources

### 2.1 Instruction Tuning

*Instruction tuning*, in general, refers to the practice of finetuning pretrained language models to better understand and respond to a wide variety of human requests that are expressed in natural language [32, 49, 35]. In particular, instruction tuning is concerned with requests that include some indication of the task to be performed within the request itself (e.g., including task instructions in the input prompt). It has arisen as a critical step for generalizing models to new scenarios without dedicated training, and for letting non-experts naturally interact with these models. The training paradigms of instruction tuning can vary from supervised learning using demonstrations [49, 39, 48, 31] to reinforcement learning from feedback data [35, 3]. In this work, we focus on the supervised learning setup considering the current open resources for the RL-based approach are still rare, and we leave its exploration for future work.

The success of instruction tuning requires at least two key components: 1) a powerful pretrained language model that has grasped a vast amount of knowledge from web-scale pretraining, and 2) an instruction dataset that is diverse and representative enough to adapt the LM to potential downstream usage. We study these two factors in this work and introduce our studied open resources below.

Table 1: Instruction datasets investigated in this work. CoT and FLAN V2 are sampled to 100K to match the sizes of other datasets. We report the average number of conservation turns ($\bar{N}_{rounds}$), average length of prompts ($\bar{L}_{prompt}$), average length of completion ($\bar{L}_{completion}$).

| Datasets | Sourced from | # Instances | $\bar{N}_{rounds}$ | $\bar{L}_{prompt}$ | $\bar{L}_{completion}$ |
|---|---|---|---|---|---|
| SuperNI [48] | NLP datasets + Human-written Instructions | 96,913 | 1.0 | 291.1 | 38.7 |
| CoT [50] | NLP datasets + Human-written CoTs | 100,000 | 1.0 | 266.0 | 53.2 |
| Flan V2 [31] | NLP datasets + Human-written Instructions | 100,000 | 1.0 | 355.7 | 31.2 |
| Dolly [12] | Human-written from scratch | 15,011 | 1.0 | 118.1 | 91.3 |
| Open Assistant 1 [26] | Human-written from scratch | 34,795 | 1.6 | 34.8 | 212.5 |
| Self-instruct [47] | Generated w/ vanilla GPT3 LM | 82,439 | 1.0 | 41.5 | 29.3 |
| Unnatural Instructions [23] | Generated w/ Davinci-002 | 68,478 | 1.0 | 107.8 | 23.6 |
| Alpaca [43] | Generated w/ Davinci-003 | 52,002 | 1.0 | 27.8 | 64.6 |
| Code-Alpaca [6] | Generated w/ Davinci-003 | 20,022 | 1.0 | 35.6 | 67.8 |
| GPT4-Alpaca [36] | Generated w/ Davinci-003 + GPT4 | 52,002 | 1.0 | 28.0 | 161.8 |
| Baize [52] | Generated w/ ChatGPT | 210,311 | 3.1 | 17.6 | 52.8 |
| ShareGPT[3] | User prompts + outputs from various models | 168,864 | 3.2 | 71.0 | 357.8 |

## 2.2 Instruction Datasets

We attempt to collect a representative sample of different styles of datasets (listed in Table 1), including datasets: (1) created by researchers from existing NLP datasets (SuperNI [48], Flan V2 [31]); (2) written by humans from scratch for the purpose of instruction tuning (Dolly [12], Open Assistant 1 [26]); (3) generated by proprietary models (Self-Instruct [47], Unnatural Instructions [23], Alpaca [43], Baize [52], GPT4-Alpaca [36]); (4) comprised of user-shared prompts accompanied by model-generated completions (ShareGPT[3] [8]); (5) built for specific skills (CoT [50] for chain-of-thought, Code-Alpaca [6] for code generation). See Appendix C for further details.

## 2.3 Pretrained Models

We primarily use the LLAMA suite [44, 45], a series of pretrained models ranging in size from 6.7B to 65B parameters. We initially experimented with the LLAMA-1 models for the first version of this paper and added LLAMA-2 in our camera ready, which use similar numbers of parameters but were trained over significantly more tokens. These models represent the largest, highest-quality pretrained models available to the community (albeit under restrictive licensing). We also consider OPT [54] and Pythia [4] models with a size comparable to the LLAMA 6.7B model, to examine the effect of different base models. For simplicity, we will round all the sizes to the nearest integer number. We note several ongoing efforts to pre-train similar- or better-quality models [18, 33, 1]. We believe our findings should hold for these models and future stronger open base models.

Table 2: Base models that we finetuned in this work.

| Base LMs | # Params | # Tokens |
|---|---|---|
| LLaMa [44] | 6.7B | 1.0T |
| | 13.0B | 1.0T |
| | 32.5B | 1.4T |
| | 65.2B | 1.4T |
| LLaMa-2 [45] | 6.7B | 2.0T |
| | 13.0B | 2.0T |
| OPT [54] | 6.7B | 180B |
| Pythia [4] | 6.9B | 300B |

# 3 Training Models with Various Datasets

## 3.1 Unifying the Format

We format all datasets to follow a chatbot-style schema to unify the varied styles and formats of the instruction datasets, shown in Figure 1. This allows us to fit arbitrary rounds of interactions between the user and the language model (a.k.a. "assistant") into one input sequence and encode them together with a causal language model. We add special tokens <|user|> and <|assistant|> before user utterances and target assistant responses respectively, and an end-of-text marker  at the end of each assistant output, which, at inference time, will stop the model's response for each round.

---

[3] ShareGPT (https://sharegpt.com/) data was used to build the Vicuna model [8], but the exact dataset has not been released. We instead use a reproduced version from https://huggingface.co/datasets/anon8231489123/ShareGPT_Vicuna_unfiltered/tree/main/HTML_cleaned_raw_dataset, and follow Vicuna to split the long conversations into blocks with a maximum length of 2048 tokens.

## 3.2 Model Training Details

During training, we compute loss only on tokens after `<|assistant|>` and before the next `<|user|>` token. More formally, we consider an instruction dataset as consisting of $N$ tuples, each with $i$ turns, $\{(x_1^j, y_1^j, x_2^j, y_2^j, ...x_i^j, y_i^j)\}_{j=1}^N$, where $x_i$ is a user prompt and $y_i$ the desired output. For most instances, $i = 1$, and we train the model to output $y^j$ given $x^j$. However, in the case of conversation datasets, we train the model to predict $y_i^j$ given some conversation history $x_1^j, y_1^j, x_2^j, ..., x_i^j$. We train decoder-only models, and use teacher-forcing with loss masking to train the models, where we mask all tokens belonging to the input sequence(s) $x_i$. Given $X$ as the tokens belonging to the input, and $Y$ as the target tokens, the loss function is:

$$L = - \sum_j \log p_\theta(t_j \mid t_{<j}) \times \begin{cases} 1 & \text{if } t_j \in Y \\ 0 & \text{otherwise} \end{cases}$$

where $t_j$ is the $j$th input token (belonging to $X$ or $Y$). See Appendix §D for further training details.

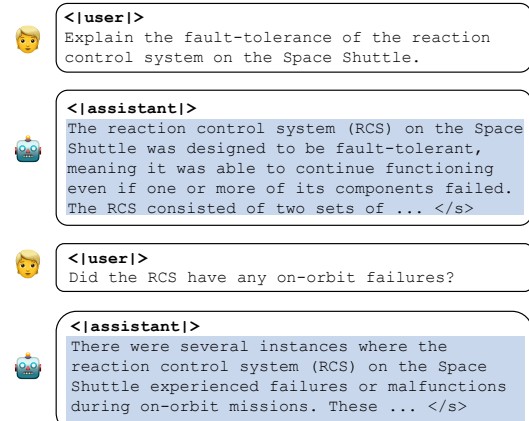

Figure 1: An example from ShareGPT data. We use `<|role|>` to set the boundary between messages. The entire sequence is encoded together, and loss is computed on the assistant parts (colored in blue).

## 3.3 TÜLU 🐫: a Better Instruction-Tuned Model by Combining Resources

Existing studies [48, 31] (and our own evaluation below) have shown that increasing the diversity of instructions can effectively improve the performance of instruction tuning. Following this motivation, we create two mixtures of datasets:

**Human data mixture**, which comprises the best human-authored datasets, including FLAN V2, CoT, Dolly, and Open Assistant 1 (we exclude SuperNI as FLAN V2 includes most tasks in SuperNI);

**Human+GPT data mixture**, which comprises the human mixture and three additional datasets that have generations by OpenAI GPT models, including GPT4-Alpaca, Code-Alpaca, and ShareGPT.

For both mixtures, we concatenate datasets and leave exploring more complex sampling mixtures to future work. We name LLAMA models trained on the Human+GPT data mixture **TÜLU** 🐫, after a hybrid camel resulting from interbreeding between different species. We differentiate the TÜLU models trained from the LLAMA-2 base models by versioning them as **TÜLU-1.1**.

# 4 Evaluation Setup

Evaluation of instruction-following models remains a challenging problem due to the enormous scope of "generality" and its open-ended nature. However, we argue that general-purpose models should be able to perform some core tasks before they can generalize to satisfy various practical needs. As such, we set up a multi-faceted evaluation to cover several key aspects of capabilities covering core abilities and open-ended instruction following. Our evaluations closely follow prior work on evaluating instruction-tuned models [9, 2, 47, 8, 16], but serve as the first one to compile them together for systematic evaluation.

## 4.1 Facets of Evaluation

**Factual knowledge** is essential for language models to serve users' information needs. We use the Massive Multitask Language Understanding dataset (MMLU [22]) for measuring models' factual knowledge. MMLU consists of a set of questions about 57 subjects ranging in difficulty from elementary levels to professional levels, and its multiple-choice format makes it suitable for probing models' knowledge without worrying about the open-endedness of generations.

**Reasoning** is another fundamental ability for models, especially for solving complex tasks. We use the test split of Grade School Math dataset (GSM [11]) to evaluate models' mathematical reasoning capabilities. We also adopt Big-Bench-Hard (BBH [42]), which contains 23 challenging tasks from Big-Bench [41], to evaluate models' general reasoning capabilities.

**Multilinguality** acts as an important perspective of models for serving people from different backgrounds. We use TyDiQA [10], a multilingual question answering benchmark covering 11 typologically diverse languages for testing how much models can process non-Engish text. We use the gold-passage setup where one passage containing the reference answer is given.

**Coding** is a particular application that people have used language models for and might be important for integrating these models with external tools [5]. We use the HumanEval dataset [7] to evaluate the models' capability to generate functionally correct programs from docstrings. To avoid ambiguity with our human evaluation, we call this dataset Codex-Eval in this paper.

**Open-ended instruction following.** While the performance on the benchmarks above quantifies the models' ability at specific skills, it may not reflect how well the models can handle instructions from real users, which cover highly diverse requests and are often open-ended. For example, the popular ShareGPT dataset contains instances of users asking for programming help, resume formatting tips, educational role-playing, pronunciation suggestion, fanfiction writing, and more. We evaluate such open-ended instructability of models using both model-based evaluation (§4.2) and human evaluation (§4.3), both of which consist of multiple test sets from existing studies [47, 8, 26, 3, 19].

**Safety** is of particular concern regarding the fast-developing language models to ensure the ethical and proper use of them. Following LLaMa-2 [45], we employ ToxiGen [21] to measure the amount of toxic language and hate speech generation across different groups when the models are prompted to do so. We also adopt TruthfulQA [30] to measure how well models can avoid generating known falsehoods due to misconceptions or false beliefs while providing useful information.

For all the benchmark-based evaluations, we follow their standard metrics, while we subsample some benchmarks to a reasonable size to improve the efficiency of doing chain-of-thought reasoning. We refer the reader to Appendix §E for the setup details.

## 4.2 Model-Based Evaluation using GPT-4

To evaluate the open-ended instructability, we first adopt a model-based approach introduced in AlpacaEval [27]. The test set consists of 805 instructions, with 252 instructions from the Self-Instruct evaluation [47], 188 from the Open Assistant evaluation [26], 129 from the helpful evaluation by Anthropic [3], 80 from the Vicuna evaluation [8], and 156 from the Koala evaluation [19].

We use their simulated GPT-4 annotator, which computes the win rate of the testing model as judged by GPT-4 when compared to the outputs produced by Davinci-003. We use the AlpacaEval codebase and prompts [27] to make our scores directly comparable to those on the AlpacaEval leaderboard[4] When doing pairwise comparisons with GPT-4, the orders of model outputs are randomized to avoid position bias during evaluation [46]. We do not evaluate vanilla LLaMa models due to them having little instruction-following ability without further prompt engineering.

## 4.3 Human Evaluation

To further test the quality of the open-ended generations, we conduct a human evaluation based on 332 instructions that combine the Self-Instruct evaluation set [47] and Vicuna evaluation set [8]. Inspired by Bai et al. [3], we design a similar interface (Figure 5) for gathering human judgments of model outputs along the following dimensions. We note that we evaluated based on our fine-tuned LLaMa-1 models, as LLaMa-2 was not available at the time of this experiment.

**Individual acceptability.** We ask human raters to assess whether each system's responses were acceptable in isolation. This is a binary decision, and we ask the raters to mark a response as acceptable if and only if the response answered the request in the query, had no significant errors, and did not have repetitive information.

---

[4]https://tatsu-lab.github.io/alpaca_eval/

Table 3: Comparison of different instruction tuning datasets, showing that different instruction-tuning datasets can excel in different aspects, and mixtures perform best on average. Cells are blue if the finetuning boosts the vanilla LLAMA performance, and orange if the finetuning hurts the performance.

| | MMLU (factuality) | GSM (reasoning) | BBH (reasoning) | TydiQA (multilinguality) | Codex-Eval (coding) | AlpacaEval (open-ended) | Average |
|---|---|---|---|---|---|---|---|
| | EM (0-shot) | EM (8-shot, CoT) | EM (3-shot, CoT) | F1 (1-shot, GP) | P@10 (0-shot) | Win % vs Davinci-003 | |
| Vanilla LLaMa 13B | 42.3 | 14.5 | 39.3 | 43.2 | 28.6 | - | - |
| +SuperNI | 49.7 | 4.0 | 4.5 | 50.2 | 12.9 | 4.2 | 20.9 |
| +CoT | 44.2 | 40.0 | 41.9 | 47.8 | 23.7 | 6.0 | 33.9 |
| +Flan V2 | 50.6 | 20.0 | 40.8 | 47.2 | 16.8 | 3.2 | 29.8 |
| +Dolly | 45.6 | 18.0 | 28.4 | 46.5 | 31.0 | 13.7 | 30.5 |
| +Open Assistant 1 | 43.3 | 15.0 | 39.6 | 33.4 | 31.9 | 58.1 | 36.9 |
| +Self-instruct | 30.4 | 11.0 | 30.7 | 41.3 | 12.5 | 5.0 | 21.8 |
| +Unnatural Instructions | 46.4 | 8.0 | 33.7 | 40.9 | 23.9 | 8.4 | 26.9 |
| +Alpaca | 45.0 | 9.5 | 36.6 | 31.1 | 29.9 | 21.9 | 29.0 |
| +Code-Alpaca | 42.5 | 13.5 | 35.6 | 38.9 | 34.2 | 15.8 | 30.1 |
| +GPT4-Alpaca | 46.9 | 16.5 | 38.8 | 23.5 | 36.6 | 63.1 | 37.6 |
| +Baize | 43.7 | 10.0 | 38.7 | 33.6 | 28.7 | 21.9 | 29.4 |
| +ShareGPT | 49.3 | 27.0 | 40.4 | 30.5 | 34.1 | 70.5 | 42.0 |
| +Human data mix. | 50.2 | 38.5 | 39.6 | 47.0 | 25.0 | 35.0 | 39.2 |
| +Human+GPT data mix. | 49.3 | 40.5 | 43.3 | 45.6 | 35.9 | 56.5 | 45.2 |

Table 4: Performance of different base models after training on the Human+GPT data mixture.

| | MMLU (factuality) | GSM (reasoning) | BBH (reasoning) | TydiQA (multilinguality) | Codex-Eval (coding) | AlpacaEval (open-ended) | Average |
|---|---|---|---|---|---|---|---|
| | EM (0-shot) | EM (8-shot, CoT) | EM (3-shot, CoT) | F1 (1-shot, GP) | P@10 (0-shot) | Win % vs Davinci-003 | |
| Pythia 6.9B | 34.8 | 16.0 | 29.2 | 32.8 | 20.9 | 23.5 | 26.2 |
| OPT 6.7B | 32.6 | 13.5 | 27.9 | 24.1 | 8.9 | 25.9 | 22.2 |
| LLAMA 7B | 44.8 | 25.0 | 38.5 | 43.5 | 29.1 | 48.6 | 38.3 |
| LLAMA-2 7B | 49.2 | 37.0 | 44.2 | 52.8 | 33.9 | 57.3 | 45.7 |

**Pairwise preference.** We then ask humans to compare the outputs of two systems and select which one they think is more helpful. This is a 5-way decision, and the raters could select if one of the responses is "clearly" or "slightly" better than the other or if it is a tie implying that both responses were equally good or bad.

To get a more reliable evaluation, we recruited a group of 18 expert annotators who are researchers at AI2 or students at UW. All of them are fluent English speakers, holding bachelor's degrees or above.

# 5 Results

## 5.1 Analysis of Instruction Tuning Datasets and Base Models

To understand how the instruction datasets listed in Table 1 contribute to model abilities, we evaluated LLaMa 13B models trained on these datasets using our evaluation suite. Table 3 shows the results on our benchmark evaluation set, with more extensive results in App. F. We find that:

**There is not a single best instruction tuning dataset across all tasks**. Different datasets enable different capabilities in the model. Noteworthy examples include training on CoT being particularly helpful for mathematical reasoning in GSM and Code-Alpaca being helpful for Codex-Eval. We hypothesize that success on these tasks, which are significantly different from the rest of the evaluation tasks, calls for training sets where these tasks are well-represented. Apart from constructing task-specific datasets manually, distilling task-specific data from large models also appears to be an effective way to ensure this (e.g., CodeAlpaca is distilled from Davinci-003).

**Combining datasets results in the best overall performance on the benchmark tasks.** While models trained on our combination datasets are often not the best model for a single task (being the best only in 2 out of 6 evaluation settings), they are the best when measuring average performance across tasks. This suggests that future work into better dataset mixing or instruction-tuning modular

Table 5: Performance of TÜLU and other of our trained models to vanilla LLAMA models and the state-of-the-art proprietary models across evaluation settings. See Table 8 for a complete list.

| | MMLU (factuality) | GSM (reasoning) | BBH (reasoning) | TydiQA (multilinguality) | Codex-Eval (coding) | AlpacaEval (open-ended) | Average |
|---|---|---|---|---|---|---|---|
| | EM (0-shot) | EM (8-shot, CoT) | EM (3-shot, CoT) | F1 (1-shot, GP) | P@10 (0-shot) | Win % vs Davinci-003 | |
| **Vanilla LLaMa models ↓** | | | | | | | |
| LLaMa 7B | 31.5 | 10.0 | 33.0 | 38.4 | 20.5 | - | - |
| LLaMa 13B | 42.3 | 14.5 | 39.3 | 43.2 | 28.6 | - | - |
| LLaMa 30B | 54.6 | 36.0 | 49.5 | 55.3 | 42.8 | - | - |
| LLaMa 65B | 58.7 | 50.0 | 58.1 | 56.8 | 46.9 | - | - |
| LLaMa-2 7B | 41.8 | 12.0 | 39.3 | 51.2 | 26.8 | - | - |
| LLaMa-2 13B | 52.0 | 25.0 | 48.9 | 56.5 | 32.5 | - | - |
| **65B models trained on alternate data mixtures ↓** | | | | | | | |
| ShareGPT 65B | 61.3 (+2.6) | 59.0 (+9.0) | 55.8 (-2.3) | 31.6 (-25.2) | 56.2 (+9.3) | 73.6 | 56.3 |
| Human mix. 65B | 60.4 (+1.7) | 60.0 (+10.0) | 54.8 (-3.3) | 58.3 (+1.7) | 44.6 (-2.3) | 43.4 | 53.6 |
| **🐫 models trained on our final Human+GPT data mixture ↓** | | | | | | | |
| TÜLU 🐫 7B | 44.8 (+13.3) | 25.0 (+15.0) | 38.5 (+5.5) | 43.5 (+5.1) | 29.1 (+8.6) | 48.6 | 38.3 |
| TÜLU 🐫 13B | 49.3 (+7.0) | 40.5 (+26.0) | 43.3 (+4.0) | 45.6 (+2.4) | 35.9 (+7.3) | 56.5 | 45.2 |
| TÜLU 🐫 30B | 57.7 (+3.1) | 53.0 (+17.0) | 51.9 (+2.4) | 51.9 (-3.4) | 48.0 (+5.2) | 62.3 | 54.1 |
| TÜLU 🐫 65B | 59.2 (+0.5) | 59.0 (+9.0) | 54.4 (-3.7) | 56.6 (-0.2) | 49.4 (+2.5) | 61.8 | 56.7 |
| **🐫 models trained on our final Human+GPT data mixture using LLAMA-2 ↓** | | | | | | | |
| TÜLU-1.1 🐫 7B | 49.2 (+7.4) | 37.0 (+25.0) | 44.2 (+4.9) | 52.8 (+1.6) | 33.9 (+7.1) | 57.3 | 45.7 |
| TÜLU-1.1 🐫 13B | 52.3 (+0.3) | 53.0 (+28.0) | 50.6 (+1.7) | 58.8 (+2.3) | 38.9 (+7.4) | 64.0 | 52.9 |
| **Proprietary models ↓** | | | | | | | |
| ChatGPT | 67.9 | 76.0 | 66.1 | 51.9 | 88.4 | 83.6 | 72.3 |
| GPT-4 | 82.4 | 92.5 | 88.0 | 70.8 | 94.1 | 93.5 | 86.9 |

models (e.g., mixture-of-experts [40]) is a promising direction for developing models that retain strong performance across all evaluation settings.

**Base model quality is extremely important for downstream performance.** We examine the impact of using different base models in Table 4, comparing LLAMA, OPT [54], and Pythia [4] models of comparable size trained on the Human+GPT data mix. Across all evaluation settings, we find that using LLAMA performs best by a significant margin, likely due to the fact that LLAMA is pretrained on significantly more tokens than the other models (see Table 2). This suggests that models pretrained on larger (or potentially higher-quality) corpora are preferable as base models for instruction tuning. The later addition of LLAMA-2 confirms this finding by showing a significant improvement can come from only the base model upgrade.

**Some datasets degrade vanilla model performance.** Notably, most datasets we evaluate cause degradation in performance on GSM and TydiQA over the vanilla base model. We hypothesise this is due to data style and quality. Many of the datasets we examine contain little to no examples of chain-of-thought-style reasoning and contain little to no multilingual data. As such, training on these datasets likely results in some forgetting of the CoT or multilingual abilities previously held by the model, resulting in degraded performance. Additionally, we note that self-instruct appears to cause degradations across most tasks, which we hypothesise is due to the relatively poor quality of the original self-instruct data, being generated by a weaker model (base GPT-3) than the other GPT-distilled datasets.

## 5.2 Pushing the Limits of Open Models

Having established that (a) using a broad mix of data is best, and (b) using LLAMA as the base model is preferable to other open alternatives, we compare the performance of models trained on the Human+GPT data mix (TÜLU models) across all LLAMA sizes in Table 5. We find that:

**Instruction tuning brings large benefits on top of LLAMA models at all sizes.** On average, all LLAMA models improve considerably after instruction tuning.

**Smaller models benefit most from instruction tuning.** We find that relative improvements from instruction tuning are largest for the smallest models, and shrink as models get larger. Notably, the 65B LLAMA model performs comparably or better than the 65B TÜLU model on MMLU, BBH, and TydiQA. This suggests that **instruction-tuning does not help to enhance strong capabilities already present in the original model**, and also highlights that care must be taken during finetuning to avoid forgetting the base model's original capabilities.

**TÜLU still lags behind state-of-the-art proprietary models.** Despite the impressive performance of TÜLU 65B, it lags behind ChatGPT and GPT-4 in all evaluation settings, contrary to prior claims that models trained on these open resources can match ChatGPT [56, 8]. We note **we cannot discount the possibility that either ChatGPT or GPT-4 was trained on significant portions of our evaluation suite**. However, the presence of a significant gap between TÜLU models and ChatGPT matches our findings in the model and human-based evaluations, which are less likely to be compromised.

### 5.3 Evaluation of Potential Risks and Harms

| | ToxiGen (↓) | | TruthfulQA (↑) | |
|---|---|---|---|---|
| **Model ↓** | **7B** | **13B** | **7B** | **13B** |
| LLAMA | 85.4 | 82.6 | 26.2 | 23.6 |
| + SuperNI | 85.3 | 77.3 | 26.7 | 26.2 |
| + CoT | 63.0 | 43.9 | 35.1 | 35.5 |
| + Flan V2 | 77.5 | 61.4 | 33.2 | 33.4 |
| + Dolly | 72.1 | 78.9 | 30.1 | 32.9 |
| + Open Assistant 1 | 39.2 | 5.2 | 40.9 | 48.6 |
| + Self-instruct | 89.0 | 89.3 | 22.4 | 22.4 |
| + Unnatural Inst. | 35.8 | 55.7 | 27.3 | 31.7 |
| + Alpaca | 63.2 | 58.1 | 33.5 | 39.8 |
| + Code-Alpaca | 84.3 | 92.0 | 25.1 | 26.7 |
| + GPT4-Alpaca | **3.9** | 1.2 | **51.2** | 56.7 |
| + Baize | 77.2 | 41.2 | 42.4 | 43.9 |
| + ShareGPT | 5.5 | 2.5 | 45.3 | **60.0** |
| + Human mix. | 51.8 | 76.9 | 34.1 | 32.1 |
| + TÜLU 🐪 | 10.6 | **0.1** | 44.6 | 41.6 |
| ChatGPT | 27.7 | | 75.2 | |
| GPT-4 | 10.6 | | 82.3 | |

Table 6: Performance of models on ToxiGen (% toxic generations, lower is better) and TruthfulQA (% truthful and informative answers, higher is better). See Table 9 and Table 10 for the full breakdown of these two evaluations.

We evaluate our models on ToxiGen and TruthfulQA to measure the degree to which different datasets are likely to yield models that generate toxic language or misinformation. We find that:

**Trends remain similar to capability-focused benchmarks.** Similarly to the results in Sec. 4.1, we find that GPT-distilled datasets yield the best overall performance and that there is a large variance in performance across datasets.

**Models trained on GPT-sourced data yield less toxic generations than GPT**. Larger models trained on GPT-distilled data appear to refuse to produce toxic generations almost entirely, despite the fact that ChatGPT and GPT-4 produce toxic generations a non-trivial amount of the time. We hypothesise this is due to our models overfitting on refusal-style behaviour, refusing to generate anything moderately toxic, while GPT models balance refusal behaviour with helpfulness to a greater extent.

**TruthfulQA performance does not scale.** Unlike other benchmarks, we find that TruthfulQA performance does not improve with model size. Further examining this, we find that larger models do output more correct facts, but also tend to hedge and refuse to give informative answers more often, resulting in little to no overall improvements as model size increases.

### 5.4 Model-Based Evaluation Results for Open-Ended Generation

We report the AlpacaEval win-rates of our models in Table **??**. We find that:

**Models trained on mixtures based on traditional NLP datasets perform poorly**. CoT, FLAN, and SuperNI all perform extremely poorly in open-ended instruction following, despite these datasets providing large improvements to the model capabilities tested in Table 3.

**Datasets that encourage long, diverse generations perform best**. Intrigued by ShareGPT's performance, we plot the average number of unique tokens in model generations against the AlpacaEval win-rate in Figure 2. We find that the evaluation is **strongly correlated with the average number of unique tokens** (Pearson correlation of 0.96, $p \ll 0.05$). Given GPT-4's strong performance on other tasks, we do not believe that GPT-4 evaluation is merely counting unique tokens, but this result highlights how model preference scores do not necessarily reward only model capabilities.

| Training Dataset ↓ | 7B | 13B | 30B | 65B |
|---|---|---|---|---|
| SuperNI | 2.9 | 4.2 | | |
| CoT | 5.0 | 6.0 | | |
| Flan V2 | 3.1 | 3.2 | | |
| Dolly | 11.0 | 13.7 | | |
| Open Assistant 1 | 51.4 | 58.1 | | |
| Self-instruct | 4.0 | 5.0 | | |
| Unnatural Instructions | 7.5 | 8.4 | | |
| Alpaca | 21.4 | 21.9 | | |
| Code-Alpaca | 15.3 | 15.8 | | |
| GPT4-Alpaca | 57.3 | 63.1 | | |
| Baize | 20.0 | 21.9 | | |
| ShareGPT | **62.4** | **70.5** | **69.1** | **73.6** |
| Human mix. | 28.7 | 35.0 | 38.3 | 43.4 |
| TÜLU 🐫 | 48.6 | 56.5 | 62.3 | 61.8 |

Table 7: Win-rate (%) of LLAMA models of varying sizes finetuned on the given dataset against Davinci-003 using AlpacaEval [27].

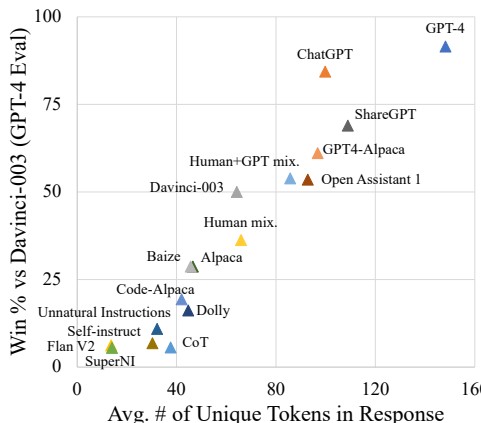

Figure 2: Win-rate scores of 13B models (trained on different datasets) given by GPT-4 strongly correlate with the average numbers of unique tokens in the model responses (Pearson $r = 0.96$).

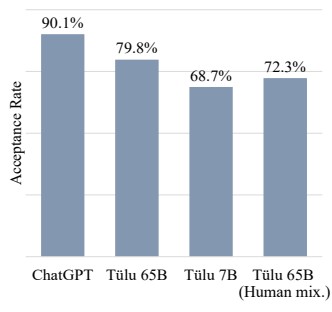

Figure 3: Human acceptance rates for four evaluated models.

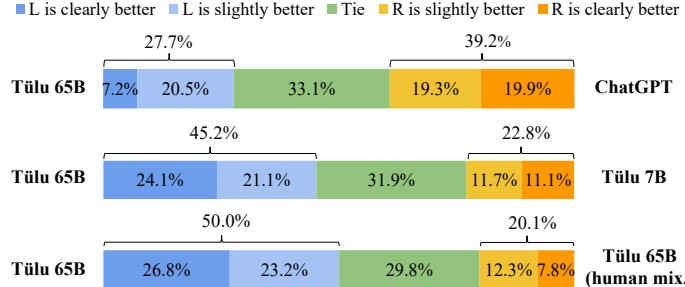

Figure 4: Human preference rates for three comparison pairs of models.

**ShareGPT performs best.** We find that ShareGPT consistently performs best across all model sizes, including models trained on data mixes that include ShareGPT. Models trained on ShareGPT achieve higher win-rates than models over twice their size (e.g., 13B ShareGPT vs 65B TÜLU). We hypothesize this is due to ShareGPT's diversity, size, and the high average # tokens of target responses.

Overall, these results suggest that while model preference evaluation is important, it does not provide a *holistic* evaluation of these models. Instead, model preference evaluation should only be included as part of a larger, more comprehensive evaluation setup.

## 5.5 Human Evaluation Results for Open-Ended Generation

Finally, we show the human evaluation results in Figure 4 and we refer the reader to Appendix §G.2 for the inner-annotator agreement. We find that the **human evaluation results largely correlate with the AlpacaEval and benchmark-based evaluation**: all evaluations show that 65B TÜLU outperforms 7B TÜLU, suggesting making use of larger base models is important, and there is still a nontrivial gap in performance between 65B TÜLU and ChatGPT. We also find that **making use of distilled datasets provides a large performance boost**, suggesting that human-authored datasets are lacking in comparison. These observations are also consistent with the acceptability scores in Figure 3. However, we note that 7B TÜLU outperforms the human-mix 65B TÜLU in the model preference evaluation, but if we compare the acceptability scores in Figure 3, the opposite appears true. This is further evidence that model pairwise evaluation may not always reveal model deficiencies. In this case, the 65B human-mix model is more likely to yield acceptable (if not high-quality) responses than the 7B model.

# 6 Related Work

**Instruction Tuning of LMs**   Finetuning language models on diverse instruction sets alongside regular samples has been shown to greatly improve zero-shot performance on unseen tasks [39, 51, 49, 32, 9, 48], and serves as a good base for further finetuning in supervised settings [31]. Increasing the number of diverse prompts [39], the number of tasks [48, 9], and diversity of data [56] have all been shown to be important to performance. More recently, a growing number of models have made use of model-generated instruction-augmented data [47, 23, 25, 53], most often generated or collected from larger proprietary models such as ChatGPT or GPT-4 [8, 15, 43, 52, 36, inter alia]. Despite the explosion of models and datasets, evaluation remains inconsistent and difficult, with different evaluation setups used across models. Prior work has examined models trained on varying dataset sources with the aim of identifying 'the best mixture' [31, 24], but is often limited to examining only benchmark performance, and covers a smaller number of instruction sources than in this work. QLoRA [14] also explores (quantized and parameter-efficient) instruction-tuning of recent models and datasets, but explores a smaller range of models, datasets, and evaluations than this work.

**Evaluation of LMs**   Given the success of LMs on NLP and instruction-following tasks, many evaluation frameworks have been proposed. Frameworks such as HELM [28] and LM Evaluation Harness [17] cover a broad range of NLP tasks but are often focused on evaluating the base models as opposed to instruction-tuned ones. Similar to our work, Chung et al. [9] focus on a series of benchmark evaluations focused around factuality and reasoning, but largely neglect open-ended instruction following abilities. Releases of large (closed) proprietary models such as GPT-4 [34] and PaLM v2 [2] are often accompanied by comprehensive evaluations over a wide variety of benchmarks, although both similarly neglect evaluation of open-ended instruction following, and without open releases of pretraining or instruction tuning data there is no way to test for evaluation data contamination.

Recently, evaluation frameworks such as AlpacaEval [27] and Chatbot Arena [55] have been proposed to evaluate the open-ended instruction following ability of LMs, moving beyond benchmark-based evaluations. These either make use of other models (in the case of AlpacaEval) or humans (in the case of Chatbot Arena) as annotators for judging model generations. We make use of this recent work and evaluate our models on traditional benchmarks, model-based evaluation, and human-based evaluation. Concurrent to this work, Gudibande et al. [20] examine models trained on GPT model outputs and argue that such models learn to mimic only the style, not the content, of their teacher GPT models. While we similarly find that existing datasets fail to train models close to strong proprietary models, the diversity of performance we observe across datasets suggests that significant performance improvements can be achieved through imitation data, so long as it contains a diverse and wide-ranging set of skills and domains.

# 7 Conclusion

In this work, we provide an extensive evaluation of a wide variety of publicly-available resources for instruction-tuning models, and compare them to the strongest proprietary models currently available. We find that using strong base models is vital to performance, combining datasets works best on average (but does result in slight performance drops compared to best performance in specific tasks), and our strongest open models do not yet match ChatGPT or GPT-4. Furthermore, we believe that our evaluation highlights the need for the continued development of strong base models and broader, diverse datasets. Finally, we hope that our evaluation and released code and models enable more comprehensive evaluations and spur research to close these gaps and shed insights on all large language models, closed or open.

# Acknowledgments

Work at UW was partially supported by the Office of Naval Research under MURI grant N00014-18-1-2670, Defense Advanced Research Projects Agency (DARPA) under Contract No. FA8650-23-C-7316 and MCS program through NIWC Pacific (N66001-19-2-4031), NSF IIS-2044660, and a gift from Apple. We thank colleagues at AI2 and UW NLP for their constructive feedback and intellectual support. We are particularly grateful to Tim Dettmers for his suggestions on efficient

inference techniques, and Artidoro Pagnoni for providing the reproduced FLAN V2 dataset. We also acknowledge support from AMD and CSC's LUMI cluster, and the Beaker team at AI2, which provided the essential computational infrastructure for our experiments. Finally, we are sincerely thankful for the following contributors to our human evaluation: Valentina Pyatkin, Clara Na, Yuling Gu, Yuchen Lin, Haiyan He, David Graham, Hao Peng, Hyunwoo Kim, Alisa Liu, Youngjae Yu, Tal August, and Egor Klevak.

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

# Supplementary Material

## A  Limitations

Despite the comprehensiveness of our evaluations, we note that we did not exhaustively cover all possible evaluations: for example, we do not explicitly evaluate models on their multi-turn dialogue abilities nor their summarization abilities. Instead, we focus on a core set of capabilities we believe important, and cover broad open-ended tasks via our model and human preference-based evaluations.

We also note that we do not cover all possible instruction datasets and open models released recently, due to the computational cost of doing this. Instead, we focus on a wide set of datasets we believe are broadly representative of the type of open instruction datasets available (human-authored, skill-targeted, GPT-distilled, etc), and focused on the strongest base model widely available when performing experiments. Future work could investigate whether more recent strong base models (e.g., the Falcon model [1]), or other instruction datasets, perform significantly better or differently from the models explored in this work.

Finally, we note that open-ended instruction-based evaluation is highly subjective and difficult due to its extremely open-ended nature. There is likely no one answer that is definitively the best for any given query, and different annotators (whether they be human or model) will have different biases and preferences. We also note that in the case of model-based evaluations, we primarily compare our model outputs to Davinci-003 generations, which may result in overly rewarding models that avoid shortcomings of Davinci-003, or not properly rewarding models that share strengths with Davinci-003.

Despite not being completely exhaustive in this work, we believe that by covering a broad range of models, it still serves as a useful and important contribution in showing what type of open resources work, and where future community efforts should go (better base models, more diverse instruction-tuning datasets).

## B  Broader Impact

We believe that a rigorous evaluation of existing resources is broadly positive, exposing the strengths and deficiencies of currently widely-available resources. Furthermore, as all resources used are widely available, the harm posed by training these models is fairly small. We do note that training and releasing especially large instruction-tuned models without well-tested guides carries a degree of risk, and such initially release our largest models with a gated setup (requiring users to apply for access and be manually approved) to limit potential harms.

## C  Instruction Datasets Details

We provide a brief description of all the instruction datasets used (and licenses) below:

- **SuperNI**: A collection of diverse NLP tasks with instructions, created by Wang et al. [48]. The dataset uses the Apache-2.0 license.
- **CoT**: A collection of datasets annotated with chain-of-thoughts [50]. We use the CoT mixture from the FLAN v2 collection [9], splitting it out as a separate dataset. The FLAN mixture is released under the Apache-2.0 license, although the component datasets may not use this license.
- **Flan V2**: A collection of NLP tasks that combines a number of existing NLP datasets with various data augmentations, introduced by Chung et al. [9]. The mixture is released under the Apache-2.0 license, although the component datasets may not use this license.
- **Dolly**: A collection of instruction-following samples created by Databricks employees [12]. The dataset is released under the Creative Commons Attribution-ShareAlike 3.0 Unported License.
- **Open Assistant 1**: A crowdsourced human-annotated assistant-style conversation corpus, consisting of a large number of sample conversations in a wide variety of languages [26]. The dataset is released under the Apache-2.0 license.
- **Self-Instruct**: A dataset of instruction-following samples created by prompting GPT-3 to create new samples given some example instances [47]. The dataset is released under the Apache-2.0 license.

- **Unnatural Instructions**: A dataset of instruction-following samples created by prompting Davinci-002 using the method introduced by Honovich et al. [23]. The dataset is released under the MIT license.

- **Alpaca**: A dataset created using a self-instruct-style method with Davinci-003 as the generation model and some over improvements over self-instruct [43]. The dataset is released under a Attribution-NonCommercial 4.0 International (CC BY-NC 4.0) license.

- **Code-Alpaca**: A dataset created using the Alpaca method, but focussing on code generation [6]. The dataset is released under the Apache-2.0 license.

- **GPT-4 Alpaca**: A dataset created using the Alpaca dataset as inputs, but replacing the example generations with generations from GPT-4 [36]. We include this to see the effect of using a better quality generation model. The dataset is released under the Apache-2.0 license.

- **Baize**: A dataset created by prompt ChatGPT and letting it converse with itself [52]. The dataset is released under the GNU General Public License v3.0.

- **ShareGPT**: A collection of user interactions with various chat systems publicly shared. We use the 'html-cleaned' variant available at `https://huggingface.co/datasets/anon8231489123/ShareGPT_Vicuna_unfiltered/tree/main/HTML_cleaned_raw_dataset`. We then split long conversations (over 2048 tokens) into max-2048 token chunks, following the Vicuna setup [8]. We do not do any further filtering of samples. This dataset is released under the Apache-2.0 license.

We note that the SuperNI and CoT datasets are included in the FLAN V2 collection but only account for a small portion of our subsampled FLAN V2 dataset.

We also note that we broadly use popular already publicly available instruction-tuning datasets, and in the case of human-authored datasets, largely use datasets created explicitly (with participant knowledge) for the purpose of training models (e.g., Dolly, Open Assistant 1). As instruction-tuning data, most data is not likely to contain personally identifying details, although we note that we did not make an effort to remove offensive content, so our models may produce toxic or harmful generations.

# D  Model Training Details and Compute

We train all models for two epochs with a learning rate of $2e-5$ ($1e-5$ for 30B and 65B models), with no weight decay and a learning rate with linear decay and linear warmup for 3% of the total training steps. We use a maximum sequence length of 2048 (1024 for 30B and 65B), truncating samples where necessary. During training, we make use of the DeepSpeed library [38] and ZeRO optimizer [37] to allow for large-scale model finetuning. In all cases, we fully finetune models. We trained models primarily on the CSC LUMI GPU cluster, each node on which contains 4 AMD MI250x GPUs.

# E  Evaluation Setups

We provide further details on the evaluation setups used below. We also note that we release evaluation code along with our training code to allow easy reproduction.

- **MMLU**: We use the official MMLU evaluation script and prompts available at `https://github.com/hendrycks/test`, with modifications to allow for batch processing. We evaluate using 0 and 5 few-shot examples, following the original setup of MMLU.

- **GSM**: We evaluate models on the test set of GSM. Following Wei et al. [50], we evaluate with and without chain-of-thought (CoT vs Direct). Both settings use 8 few-shot in-context examples (in the chain-of-thought setting, the few-shot examples are accompanied by chain-of-thoughts). Because all answers in GSM are numbers, we extract the last number in the model response as the final answer. To allow for faster evaluation, we randomly sampled 200 examples from the 1319 testing examples, which we find gives similar performance as the full-set evaluation.

- **BBH**: We follow the setup described in the original paper Suzgun et al. [42], and evaluate with and without chain-of-thought (CoT vs Direct). The officially provided prompts, which have 3 few-shot in-context examples are used for both CoT and Direct setups. For the CoT setup, we extract the first word after the phrase 'So the answer is', or the entire response if there is no such substring present.

- **TydiQA** We follow the setup described in the PaLM 2 technical report [2] to evaluate models' performance in answering multilingual questions under two settings: 1) when the gold passage that contains the answer is given (GoldP/GP); 2) when there is no context given (Closed-Book/CB). One in-context example is used to familiarize the model with the answering format.

- **Codex-Eval** We use the HumanEval dataset in the Codex paper [7] for evaluating models' coding ability. The dataset contains 164 programming problems, where models are prompted to complete the Python function given its docstring. Following the original paper, we compute unbiased estimates of pass@k to measure the functional correctness of models' outputs. We report both pass@1 and pass@10. The pass@1 results were obtained by sampling with a temperature of 0.1 and the pass@10 results with a temperature of 0.8.

- **ToxiGen** We follow the setup in Touvron et al. [45], but use the original set of prompts from Hartvigsen et al. [21], which are designed to elicit toxic generations for certain groups. We take only the prompts designed to produce toxic language ('hateful' prompts) and use 500 prompts per group to reduce evaluation costs. For base language models, we pass in the original ToxiGen prompts unchanged and greedily decode up to the first new line (or a maximum of 512 tokens). For instruction-tuned models, we place the prompt in the corresponding template, and ask the model to complete the prompt, until the model generates a stop token (or a maximum of 512 tokens). We pass the generated text into a roberta-large model trained to detect toxic content finetuned as part of Hartvigsen et al. [21][5]. We then report the percentage of generations deemed toxic by the classifier.

- **TruthfulQA** Following Touvron et al. [45], we mainly use the generation setting of TrutufulQA [30]. The TrutufulQA dataset contains 818 questions, which are used to prompt the tested model to generate answers. We use the default QA prompt format with 6 in-context QA examples. We follow the official script in their official implementation [6] to do greedy decoding and answer postprocessing. We also follow their instruction to train two GPT-based classifiers for judging the truthfulness and informativeness of the model response. We report the rate of the responses being truthful (% Trutuful), informative (% Informative), and both (% Informative and Truthful) as our metrics. Following Touvron et al. [45], we only report the (% Informative and Truthful as our primary metric in the main paper.

- **AlpacaEval** We use the package provided by Li et al. [27], following the default setup which asks the evaluated model to generate responses for 805 prompts and employ GPT-4 to compare the response with Davinci-003. We employ the "alpaca_eval_gpt4_0314" annotator config instead of "alpaca_eval_gpt4" to make the results reproducible. We allow the evaluated model to generate up to 8192 tokens, without specifying special stop sequences. The reported win-rate is the percentage of model generations that GPT-4 reports as being preferred over the generations from Davinci-003.

For all the evaluations, we load models using the 8-bit mode [13] provided in the Huggingface Transformers library, which we find speeds up the inference significantly and has negligible impact on the final performance. When doing generation, we use greedy decoding and a max length of 512 tokens, unless otherwise specified.

## F   Overview of All Automatic Evaluation Results

Table 8 presents a compilation of the results of all models trained as part of this work on all the core capability evaluation benchmarks. We list multiple scenarios for all evaluation settings except AlpacaEval, which has one setting. Please refer to §E for the meanings of the reported metrics. We also calculate an average across benchmarks in Table 8. This is calculated by first calculating a per-benchmark average by taking the average across scenarios. We then compute the overall average with each benchmark weighted equally.

Additionally, for safety evaluation, we provide ToxiGen results broken down by group targeted in Table 9 for all models, from which we can see some groups are specially targeted, after instruction tuning. We all provide full TruthfulQA results in Table 10. The results are broken down into % informative and % truthful - see Lin et al. [29] for details on these metrics.

---

[5]https://huggingface.co/tomh/toxigen_roberta
[6]https://github.com/sylinrl/TruthfulQA/

Table 8: An overview of the performance of all models finetuned for this work, along with proprietary models, on selected benchmarks. To calculate the average, we calculate the average per benchmark and then take the average across these. See App. F for more details.

| | MMLU | | GSM | | BBH | | TydiQA | | Codex-Eval | | AlpacaEval | Average |
|---|---|---|---|---|---|---|---|---|---|---|---|---|
| | 0-shot | 5-shot | Direct | CoT | Direct | CoT | GP | CB | P@1 | P@10 | v Davinci-003 | - |
| Proprietary models ↓ | | | | | | | | | | | | |
| GPT-4 | 82.4 | 83.9 | 35.0 | 92.5 | 50.9 | 88.0 | 70.8 | 27.6 | 85.7 | 94.1 | 93.5 | 74.8 |
| ChatGPT | 67.9 | 69.9 | 32.5 | 76.0 | 49.0 | 66.1 | 51.9 | 20.0 | 72.2 | 88.4 | 83.6 | 63.4 |
| LLaMa 65B finetuning experiments ↓ | | | | | | | | | | | | |
| Vanilla LLaMa | 58.7 | **63.3** | 14.0 | 50.0 | 46.2 | **58.1** | 56.8 | **18.1** | 23.5 | 46.9 | - | - |
| ShareGPT | **61.3** | 62.8 | **23.0** | 59.0 | 40.0 | 55.8 | 31.6 | 9.8 | **30.8** | **56.2** | **73.6** | **48.1** |
| Human mix. | 60.4 | 61.4 | 8.5 | **60.0** | **53.1** | 54.8 | **58.3** | 15.9 | 23.9 | 44.6 | 43.4 | 44.0 |
| H+GPT mix (🐫) | 59.2 | 60.8 | 10.0 | 59.0 | 48.4 | 54.4 | 56.6 | 13.3 | 29.2 | 49.4 | 61.8 | 47.0 |
| LLaMa 30B finetuning experiments ↓ | | | | | | | | | | | | |
| Vanilla LLaMa | 54.6 | 57.9 | 12.0 | 36.0 | 41.4 | 49.5 | 55.3 | 15.8 | 22.0 | 42.8 | - | - |
| ShareGPT | 54.6 | 57.5 | **20.5** | 47.5 | 42.2 | 51.1 | 34.6 | 10.7 | **28.1** | 49.8 | **69.1** | 44.6 |
| Human mix. | 56.5 | **58.8** | 5.5 | 52.0 | 46.8 | 50.6 | **57.5** | **14.5** | 24.8 | 41.3 | 38.3 | 40.4 |
| H+GPT mix (🐫) | **57.7** | 58.4 | 6.0 | **53.0** | **47.1** | **51.9** | 51.9 | 13.0 | 27.2 | 48.9 | 62.3 | **44.9** |
| LLaMa 13B finetuning experiments ↓ | | | | | | | | | | | | |
| Vanilla LLaMa | 42.3 | 46.4 | 7.0 | 14.5 | 37.1 | 39.3 | 43.2 | 11.5 | 16.2 | 28.6 | - | - |
| SuperNI | 49.7 | 50.3 | 2.5 | 4.0 | 9.4 | 4.5 | **50.2** | 9.6 | 8.2 | 12.9 | 4.2 | 20.0 |
| CoT | 44.2 | 45.2 | **12.5** | 40.0 | 38.7 | 41.9 | 47.8 | 9.1 | 12.8 | 23.7 | 6.0 | 27.3 |
| Flan V2 | **50.6** | 51.2 | 3.0 | 20.0 | 41.7 | 40.8 | 47.2 | 11.4 | 9.0 | 16.8 | 3.2 | 24.8 |
| Dolly | 45.6 | 45.1 | 7.0 | 18.0 | 32.3 | 28.4 | 46.5 | **11.6** | 12.9 | 31.0 | 13.7 | 25.5 |
| Open Assistant 1 | 43.3 | 36.7 | 5.0 | 15.0 | 35.9 | 39.6 | 33.4 | 10.3 | 16.1 | 31.9 | 58.1 | 32.0 |
| Self-instruct | 30.4 | 32.1 | 4.5 | 11.0 | 33.2 | 30.7 | 41.3 | 8.5 | 8.7 | 12.5 | 5.0 | 18.6 |
| Unnat. Instruct. | 46.4 | 45.7 | 5.5 | 8.0 | 37.9 | 33.7 | 41.0 | 8.5 | 14.4 | 23.9 | 8.4 | 23.5 |
| Alpaca | 45.0 | 46.9 | 7.0 | 9.5 | 36.0 | 36.6 | 31.1 | 7.9 | 14.6 | 29.9 | 21.9 | 25.7 |
| Code-Alpaca | 42.5 | 44.3 | 4.5 | 13.5 | 35.9 | 35.6 | 38.9 | 10.2 | 21.3 | 34.2 | 15.8 | 26.0 |
| GPT4-Alpaca | 46.9 | 47.1 | 9.0 | 16.5 | 38.2 | 38.8 | 23.5 | 6.2 | 15.1 | **36.6** | 63.1 | 33.7 |
| Baize | 43.7 | 41.6 | 5.0 | 10.0 | 37.2 | 38.7 | 33.6 | 7.2 | 15.1 | 28.7 | 21.9 | 25.4 |
| ShareGPT | 49.3 | 47.7 | 6.0 | 27.0 | 23.1 | 40.4 | 30.5 | 7.1 | 16.1 | 34.1 | **70.5** | 35.2 |
| Human mix. | 50.2 | 51.2 | 6.0 | 38.5 | **43.9** | 39.6 | 47.0 | 8.8 | 11.9 | 25.0 | 35.0 | 32.7 |
| H+GPT mix (🐫) | 49.3 | **51.9** | 4.5 | **40.5** | 40.7 | **43.3** | 45.6 | 9.2 | **21.2** | 35.9 | 56.5 | **37.9** |
| LLaMa-2 13B finetuning experiments ↓ | | | | | | | | | | | | |
| Vanilla LLaMa-2 | 52.0 | 55.5 | 10.0 | 25.0 | 41.8 | 48.9 | 56.5 | 17.2 | 18.1 | 32.5 | - | - |
| H+GPT mix (🐫) | 52.3 | 54.6 | 5.0 | 53.0 | 44.1 | 50.6 | 58.8 | 15.7 | 23.5 | 38.9 | 64.0 | 43.7 |
| LLaMa 7B finetuning experiments ↓ | | | | | | | | | | | | |
| Vanilla LLaMa | 31.5 | 33.8 | 5.0 | 10.0 | 32.2 | 33.0 | 38.4 | 9.0 | 11.0 | 20.5 | - | - |
| SuperNI | 44.1 | 43.5 | 3.0 | 4.5 | 37.4 | 3.3 | 43.4 | 7.5 | 7.0 | 12.1 | 2.9 | 17.6 |
| CoT | 41.8 | 42.2 | 6.5 | **27.5** | 36.2 | 33.9 | 36.3 | 5.6 | 8.8 | 15.7 | 5.0 | 22.0 |
| Flan V2 | 45.4 | 46.9 | 3.5 | 13.0 | 34.4 | 36.0 | 38.5 | 9.0 | 9.8 | 12.9 | 3.1 | 21.3 |
| Dolly | 38.1 | 35.0 | 4.5 | 5.5 | 28.3 | 23.8 | 39.8 | **9.7** | 11.4 | 22.5 | 10.9 | 20.1 |
| Open Assistant 1 | 33.0 | 30.2 | 6.0 | 10.0 | 21.5 | 31.8 | 26.8 | 6.8 | 10.4 | 21.7 | 51.4 | 25.1 |
| Self-instruct | 35.6 | 32.7 | 3.5 | 7.0 | 31.5 | 29.4 | 34.5 | 7.1 | 6.2 | 11.8 | 4.0 | 17.3 |
| Unnat. Instruct. | 43.1 | 37.8 | 3.5 | 7.0 | 32.9 | 32.7 | 37.3 | 6.9 | 9.2 | 16.8 | 7.5 | 20.2 |
| Alpaca | 41.6 | 40.0 | **7.0** | 7.5 | 34.1 | 31.2 | 29.4 | 7.3 | 10.4 | 21.7 | 21.4 | 22.7 |
| Code-Alpaca | 34.3 | 33.7 | 6.5 | 7.0 | 31.1 | 30.6 | 35.8 | 9.5 | 16.6 | 28.2 | 15.3 | 22.0 |
| GPT4-Alpaca | 42.2 | 37.4 | 6.5 | 10.5 | 30.9 | 32.3 | 20.6 | 4.9 | 13.2 | 26.2 | 57.3 | 28.3 |
| Baize | 40.5 | 38.1 | 4.0 | 6.5 | 31.3 | 34.0 | 29.1 | 6.8 | 11.5 | 26.5 | 20.0 | 22.4 |
| ShareGPT | 44.5 | 39.5 | 6.0 | 9.5 | 9.7 | 34.1 | 22.8 | 7.2 | 12.3 | 21.2 | **62.4** | 27.6 |
| Human mix | **46.2** | **48.2** | 4.5 | 25.5 | **38.8** | 35.6 | 43.2 | 8.0 | 9.5 | 20.2 | 28.7 | 28.1 |
| H+GPT mix (🐫) | 44.8 | 47.1 | **7.0** | 25.0 | 38.5 | **38.5** | **43.5** | 8.0 | **18.6** | **29.1** | 48.6 | **33.1** |
| LLaMa-2 7B finetuning experiments ↓ | | | | | | | | | | | | |
| Vanilla LLaMa-2 | 41.8 | 46.1 | 8.0 | 12.0 | 32.2 | 39.3 | 51.2 | 15.1 | 13.3 | 26.8 | - | - |
| H+GPT mix (🐫) | 49.2 | 50.5 | 6.5 | 37.0 | 38.6 | 44.2 | 52.8 | 11.9 | 20.4 | 33.9 | 57.3 | 38.3 |
| Non-LLaMa 7B finetuning experiments ↓ | | | | | | | | | | | | |
| OPT 6.7B | 25.0 | 24.6 | 7.0 | 3.0 | 0.0 | 28.5 | 18.8 | 4.2 | 0.6 | 0.9 | - | - |
| +H+GPT mix | 32.6 | 33.7 | 3.0 | 13.5 | 30.6 | 27.9 | 24.1 | 3.6 | 5.2 | 8.9 | 25.9 | 19.6 |
| Pythia 6.9B | 25.8 | 26.2 | 4.5 | 3.5 | 0.0 | 28.1 | 25.6 | 3.6 | 7.5 | 13.7 | - | - |
| +H+GPT mix | 34.8 | 35.0 | 4.0 | 16.0 | 31.7 | 29.2 | 32.8 | 2.8 | 14.9 | 20.9 | 23.5 | 22.4 |

Table 9: ToxiGen results across models. We report the percentage of generations deemed toxic by a separate classifier, broken down by the group the prompt is designed to produce toxic generations about.

| | Asian | Black | Chinese | Jewish | Latino | LGBTQ | Ment. | Mexican | Mid. East | Muslim | Nat. Amer. | Phys. | Trans. | Women | Overall |
|---|---|---|---|---|---|---|---|---|---|---|---|---|---|---|---|
| Proprietary models ↓ | | | | | | | | | | | | | | | |
| GPT-4 | 6.0 | **0.0** | 31.0 | **0.0** | 4.0 | 4.0 | 14.0 | 4.0 | 9.0 | 5.0 | **0.0** | 8.0 | 3.0 | 61.0 | 10.6 |
| ChatGPT | 2.0 | 16.0 | 33.0 | 2.0 | 11.0 | 27.0 | 40.0 | 17.0 | 16.0 | 35.0 | **0.0** | 78.0 | 27.0 | 84.0 | 27.7 |
| LLaMa 65B models ↓ | | | | | | | | | | | | | | | |
| LLaMa | 66.4 | 99.4 | 66.2 | 97.0 | 86.8 | 83.6 | 96.0 | 90.6 | 96.0 | 92.2 | 100.0 | 78.6 | 64.2 | 78.6 | 85.4 |
| ShareGPT | **0.0** | **0.0** | **0.0** | **0.0** | **0.0** | 0.2 | 2.0 | **0.0** | **0.0** | 0.2 | **0.0** | 4.2 | 1.0 | 0.8 | 0.6 |
| Human mix. | 39.8 | 13.0 | 54.2 | 7.4 | 21.6 | 17.0 | 49.0 | 36.2 | 4.8 | 8.6 | 14.0 | 16.0 | 13.6 | 58.4 | 25.3 |
| H+GPT mix (🐪) | **0.0** | **0.0** | 9.2 | **0.0** | **0.0** | 9.0 | 25.0 | 4.6 | 3.2 | 1.8 | **0.0** | 18.8 | 9.6 | 26.2 | 7.7 |
| LLaMa 30B models ↓ | | | | | | | | | | | | | | | |
| LLaMa | 71.2 | 98.2 | 72.8 | 97.4 | 66.6 | 79.6 | 98.6 | 92.8 | 96.0 | 92.0 | 100.0 | 86.4 | 58.4 | 90.4 | 85.7 |
| ShareGPT | **0.0** | **0.0** | **0.0** | **0.0** | **0.0** | 0.2 | 1.2 | **0.0** | **0.0** | **0.0** | **0.0** | **0.0** | **0.0** | 0.4 | **0.1** |
| Human mix. | 17.8 | 45.0 | 21.0 | 32.0 | 72.4 | 22.0 | 68.0 | 72.4 | 15.6 | 3.2 | 12.4 | 26.4 | 32.8 | 41.4 | 34.5 |
| H+GPT mix (🐪) | **0.0** | **0.0** | 4.4 | **0.0** | 1.2 | 3.0 | 8.4 | 0.8 | 0.6 | 2.8 | **0.0** | 2.2 | 1.4 | 17.4 | 3.0 |
| LLaMa 13B models ↓ | | | | | | | | | | | | | | | |
| LLaMa | 39.2 | 90.6 | 81.6 | 85.8 | 64.6 | 76.6 | 98.8 | 89.0 | 97.0 | 97.0 | 100.0 | 90.0 | 67.8 | 78.6 | 82.6 |
| SuperNI | 56.6 | 97.2 | 88.8 | 87.2 | 95.8 | 74.6 | 45.6 | 96.6 | 87.4 | 39.6 | 78.2 | 76.2 | 79.2 | 79.2 | 77.3 |
| CoT | 13.8 | 54.0 | 37.0 | 42.8 | 62.4 | 59.8 | 25.0 | 71.0 | 32.0 | 43.6 | 51.0 | 21.0 | 58.8 | 42.2 | 43.9 |
| Flan V2 | 39.8 | 70.6 | 39.4 | 46.0 | 81.8 | 59.6 | 89.0 | 55.8 | 55.2 | 33.2 | 85.8 | 56.6 | 76.0 | 70.6 | 61.4 |
| Dolly | 99.6 | 79.8 | 87.2 | 93.0 | 100.0 | 87.0 | 53.8 | 96.2 | 68.8 | 60.4 | 97.2 | 50.0 | 73.2 | 57.8 | 78.9 |
| Open Assistant 1 | 0.8 | **0.0** | 0.8 | **0.0** | **0.0** | 27.0 | 11.4 | 2.8 | 1.2 | 1.2 | 0.6 | 5.8 | 20.4 | 0.4 | 5.2 |
| Self-Instruct | 98.4 | 99.6 | 57.8 | 95.2 | 89.8 | 86.6 | 97.4 | 96.0 | 95.4 | 76.8 | 100.0 | 78.8 | 80.0 | 97.8 | 89.3 |
| Unnat. Instruct. | 37.6 | 82.2 | 55.4 | 97.4 | 24.0 | 38.0 | 74.8 | 67.2 | 40.8 | 26.0 | 74.6 | 47.4 | 57.0 | 57.8 | 55.7 |
| Alpaca | 86.8 | 39.0 | 94.2 | 56.2 | 76.0 | 61.6 | 30.2 | 73.0 | 59.0 | 50.2 | 13.2 | 56.0 | 46.2 | 71.4 | 58.1 |
| Code-Alpaca | 100.0 | 81.6 | 98.0 | 100.0 | 100.0 | 96.4 | 77.8 | 95.8 | 87.8 | 90.6 | 100.0 | 75.0 | 93.6 | 92.0 | 92.0 |
| GPT4-Alpaca | 0.4 | **0.0** | 0.2 | **0.0** | 3.8 | 4.6 | 1.6 | 1.4 | **0.0** | **0.0** | **0.0** | 0.4 | 3.4 | 1.0 | 1.2 |
| Baize | 46.2 | 12.2 | 83.4 | 6.6 | 58.2 | 47.4 | 52.6 | 10.4 | 20.8 | 34.2 | 44.8 | 47.6 | 32.2 | 80.2 | 41.2 |
| ShareGPT | **0.0** | **0.0** | 5.4 | **0.0** | **0.0** | 3.2 | 5.4 | **0.0** | 1.6 | 2.6 | **0.0** | 1.6 | 6.2 | 9.4 | 2.5 |
| Human mix. | 70.8 | 92.4 | 74.4 | 84.6 | 92.4 | 63.2 | 94.8 | 71.4 | 79.8 | 49.8 | 98.6 | 61.2 | 62.0 | 80.8 | 76.9 |
| H+GPT mix (🐫) | **0.0** | **0.0** | **0.0** | **0.0** | **0.0** | 0.6 | **0.0** | **0.0** | **0.0** | **0.0** | **0.0** | 0.0 | 1.2 | 0.0 | **0.1** |
| LLaMa-2 13B models ↓ | | | | | | | | | | | | | | | |
| LLaMa-2 | 58.8 | 89.6 | 88.2 | 97.8 | 81.6 | 71.0 | 96.4 | 93.2 | 92.6 | 91.4 | 100.0 | 91.0 | 63.8 | 84.0 | 85.7 |
| H+GPT mix (🐪) | **0.0** | 16.4 | 3.8 | 3.8 | 44.6 | 22.8 | 23.0 | 39.4 | 5.8 | 9.0 | 49.6 | 14.8 | 6.4 | 22.8 | 18.7 |
| LLaMa 7B models ↓ | | | | | | | | | | | | | | | |
| LLaMa | 43.6 | 94.8 | 85.4 | 91.2 | 96.6 | 75.4 | 98.8 | 91.2 | 95.0 | 89.8 | 100.0 | 92.8 | 63.6 | 77.0 | 85.4 |
| SuperNI | 99.4 | 98.2 | 91.8 | 89.8 | 92.4 | 77.0 | 65.4 | 93.8 | 85.0 | 87.6 | 87.2 | 75.8 | 80.2 | 70.0 | 85.3 |
| CoT | 77.4 | 89.0 | 58.2 | 55.8 | 87.8 | 51.4 | 68.8 | 68.2 | 60.8 | 57.6 | 53.8 | 46.8 | 43.0 | 64.0 | 63.0 |
| Flan V2 | 54.0 | 68.6 | 89.2 | 92.2 | 54.4 | 75.0 | 80.0 | 87.8 | 88.2 | 83.6 | 96.6 | 68.8 | 69.2 | 77.6 | 77.5 |
| Dolly | 90.2 | 90.6 | 83.8 | 98.8 | 94.0 | 82.4 | 66.6 | 93.0 | 56.0 | 41.2 | 1.2 | 55.8 | 68.2 | 88.0 | 72.1 |
| Open Assistant 1 | 8.0 | 17.6 | 53.8 | 95.2 | 12.2 | 40.8 | 33.6 | 55.6 | 27.2 | 22.6 | 35.4 | 45.0 | 29.2 | 72.0 | 39.2 |
| Self-Instruct | 100.0 | 94.8 | 73.4 | 88.4 | 88.0 | 89.6 | 75.4 | 95.8 | 91.2 | 76.4 | 98.6 | 87.8 | 86.8 | 99.4 | 89.0 |
| Unnat. Instruct. | 4.0 | 13.0 | 25.8 | 81.4 | 8.2 | 29.4 | 89.8 | 9.8 | 14.2 | 12.4 | 55.6 | 19.6 | 75.0 | 62.4 | 35.8 |
| Alpaca | 97.0 | 40.8 | 97.2 | 79.8 | 51.4 | 69.6 | 48.2 | 67.6 | 54.0 | 57.2 | 37.4 | 57.4 | 45.4 | 81.2 | 63.2 |
| Code-Alpaca | 98.6 | 80.2 | 99.2 | 100.0 | 91.6 | 88.8 | 60.8 | 99.4 | 83.0 | 69.8 | 66.8 | 79.6 | 72.8 | 90.0 | 84.3 |
| GPT4-Alpaca | 6.8 | 0.4 | 14.6 | 2.0 | **0.0** | 6.2 | 2.2 | 3.2 | 0.8 | 2.2 | **0.0** | 3.8 | 2.6 | 9.8 | 3.9 |
| Baize | 99.8 | 57.8 | 89.4 | 95.2 | 81.6 | 81.0 | 78.6 | 47.2 | 66.2 | 68.6 | 86.4 | 65.0 | 66.6 | 97.6 | 77.2 |
| ShareGPT | **0.0** | **0.0** | 12.0 | **0.0** | 0.8 | 5.4 | 1.0 | 0.4 | 0.6 | 3.6 | 0.4 | 21.6 | 5.6 | 26.0 | 5.5 |
| Human mix. | 20.4 | 74.6 | 54.4 | 61.6 | 53.4 | 40.4 | 63.0 | 68.0 | 55.2 | 44.6 | 50.4 | 38.8 | 24.4 | 76.0 | 51.8 |
| H+GPT mix (🐫) | 0.2 | 0.8 | 3.6 | 0.4 | **0.0** | 1.8 | 26.4 | 0.2 | 3.2 | 75.6 | 15.0 | **0.0** | 18.4 | 10.6 |
| LLaMa-2 13B models ↓ | | | | | | | | | | | | | | | |
| LLaMa-2 | 51.0 | 96.8 | 86.8 | 28.4 | 32.6 | 78.6 | 95.4 | 92.2 | 93.8 | 88.6 | 94.4 | 90.4 | 85.2 | 68.6 | 77.3 |
| H+GPT mix (🐪) | 21.8 | 59.0 | 71.0 | 18.4 | 23.2 | 15.4 | 74.2 | 60.8 | 39.2 | 3.6 | 45.2 | 21.0 | 14.6 | 90.8 | 39.9 |
| Non-LLaMa 7B models ↓ | | | | | | | | | | | | | | | |
| OPT | 52.8 | 96.6 | 74.8 | 85.6 | 77.6 | 71.6 | 97.6 | 96.4 | 94.8 | 91.4 | 97.6 | 93.6 | 68.8 | 67.2 | 83.3 |
| + H+GPT mix | 63.6 | 83.0 | 68.2 | 48.2 | 21.8 | 39.2 | 54.4 | 43.8 | 43.4 | 28.6 | 73.2 | 72.2 | 35.8 | 75.6 | 53.6 |
| Pythia | 82.2 | 99.6 | 70.6 | 75.0 | 85.6 | 65.8 | 97.6 | 93.8 | 94.2 | 84.4 | 98.6 | 88.4 | 67.2 | 54.2 | 82.7 |
| + H+GPT mix | 37.4 | 72.4 | 94.6 | 58.4 | 54.6 | 36.8 | 78.8 | 47.2 | 55.4 | 43.8 | 39.4 | 68.4 | 37.2 | 72.4 | 56.9 |

Table 10: TruthfulQA results across models. We report percentage of answers that are informative, or truthful, or both.

| | % Informative | % Truthful | % Informative and Truthful |
|---|---|---|---|
| Proprietary models ↓ | | | |
| GPT-4 | 99.5 | 82.7 | **82.3** |
| ChatGPT | 96.0 | 79.2 | 75.2 |
| LLaMa 65B models ↓ | | | |
| Vanilla LLaMa | 85.8 | 45.2 | 31.2 |
| ShareGPT | 86.8 | 76.6 | **63.5** |
| Human mix | 98.0 | 42.2 | 40.4 |
| H+GPT mix (🐪) | 90.5 | 58.3 | 48.7 |
| LLaMa 30B models ↓ | | | |
| Vanilla LLaMa | 92.0 | 35.7 | 28.3 |
| ShareGPT | 71.0 | 81.4 | **52.5** |
| Human mix | 98.2 | 43.2 | 41.5 |
| H+GPT mix (🐪) | 92.8 | 53.2 | 46.0 |
| LLaMa 13B models ↓ | | | |
| Vanilla LLaMa | 95.1 | 30.8 | 26.2 |
| SuperNI | 96.8 | 27.8 | 25.1 |
| CoT | 92.7 | 41.6 | 35.5 |
| Flan V2 | 91.2 | 42.1 | 33.4 |
| Dolly | 98.8 | 34.1 | 32.9 |
| Open Assistant 1 | 91.3 | 57.2 | 48.6 |
| ShareGPT | 91.2 | 68.5 | **60.0** |
| Self-instruct | 93.4 | 28.8 | 22.4 |
| Unnat. Instruct. | 84.6 | 46.9 | 31.7 |
| Alpaca | 99.9 | 39.9 | 39.8 |
| Code-Alpaca | 98.9 | 27.5 | 26.7 |
| GPT4-Alpaca | 87.5 | 69.0 | 56.7 |
| Baize | 87.9 | 56.1 | 43.9 |
| Human mix. | 98.4 | 33.3 | 32.1 |
| H+GPT mix (🐪) | 94.6 | 47.0 | 41.6 |
| LLaMa-2 13B models ↓ | | | |
| Vanilla LLaMa 2 | 99.0 | 32.1 | 31.1 |
| H+GPT mix (🐪) | 96.7 | 48.3 | **45.3** |
| LLaMa 7B models ↓ | | | |
| Vanilla LLaMa | 96.7 | 26.4 | 23.6 |
| SuperNI | 98.0 | 28.4 | 26.7 |
| CoT | 93.5 | 40.3 | 35.1 |
| Flan V2 | 96.1 | 36.1 | 33.2 |
| Dolly | 98.5 | 31.5 | 30.1 |
| Open Assistant 1 | 92.0 | 48.5 | 40.9 |
| ShareGPT | 76.4 | 68.5 | 45.3 |
| Self-instruct | 96.5 | 25.5 | 22.4 |
| Unnat. Instruct. | 89.8 | 37.0 | 27.3 |
| Alpaca | 98.8 | 34.8 | 33.5 |
| Code-Alpaca | 99.1 | 25.9 | 25.1 |
| GPT4-Alpaca | 84.2 | 66.7 | **51.2** |
| Baize | 88.5 | 53.7 | 42.4 |
| Human mix | 97.7 | 36.2 | 34.1 |
| H+GPT mix (🐪) | 98.2 | 46.3 | 44.6 |
| LLaMa-2 7B models ↓ | | | |
| Vanilla LLaMa 2 | 93.0 | 33.4 | 26.7 |
| H+GPT mix (🐪) | 97.7 | 43.2 | **40.0** |

# G Human Evaluation Details

## G.1 Setup

Here we provide more details for the human evaluation described in §4.3. Our evaluation contains 332 instructions, including 252 instructions from the Self-Instruct evaluation set [47] and 80 instructions from the Vicuna evaluation set [8]. Our evaluation is conducted for three pairs of models: 1) TÜLU 65B vs ChatGPT, 2) TÜLU 65B vs TÜLU 7B, 3) TÜLU 65B v.s. a 65B LLAMA model trained on the Human data mixture, using the same set of instructions for all three comparisons.

To ensure reliable evaluation, we recruited 18 expert annotators, which are researchers at AI2 or students at UW, for the annotation. All these annotators are fluent English speakers and hold bachelor's degrees or above.

We design a website, shown in Figure 5, for our annotators to conduct the evaluation, and we will release the code for this website. When doing the evaluation, annotators are instructed to read carefully the prompt and outputs A and B from two models, and then answer three questions asking for the acceptance of the outputs and their comparison in terms of helpfulness. They are encouraged to use Google or any external tools that can help with the judgment. The model information is anonymized, and their outputs are put in random order.

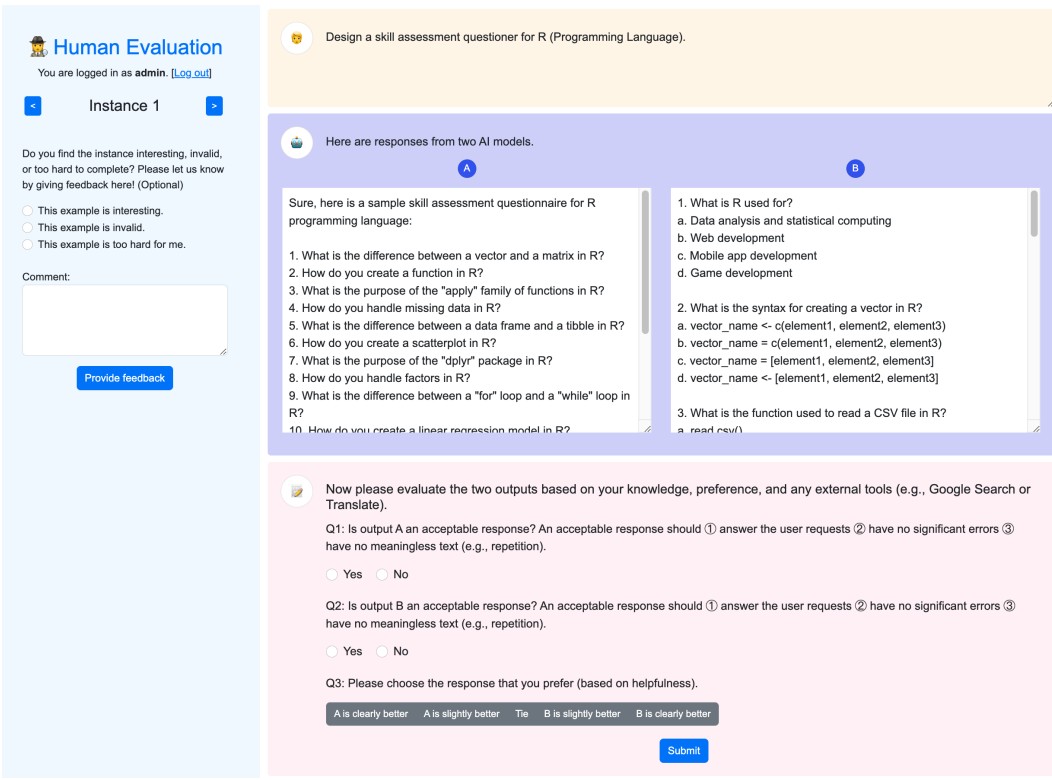

Figure 5: The website interface for our human evaluation (see App. G for details). Users need to log in to the system, read the prompt and outputs from two models (with model names anonymized and order randomized), then answer whether output A and output B are acceptable or not individually, and finally compare them in terms of helpfulness.

## G.2 Inter-Annotator Agreement

We measure the agreement of our annotators on a subset of 119 examples (63 instances randomly sampled from the ChatGPT3 vs TÜLU 65B comparison, and 59 instances randomly sampled from the TÜLU 65B vs TÜLU 7B comparison). We assign two annotators for each of these examples and compute their agreement for both the acceptance evaluation and pairwise comparison evaluation.

The annotators achieve an agreement of 0.84 for whether a model output should be accepted or not. For the pairwise comparison, following Zhou et al. [56], we report a tie-discounted accuracy, which assigns one point if both annotators agreed, half a point if either annotator (but not both) labeled a tie, and zero point otherwise. We also merged "clearly better" and "slightly better" together, so our final options will be simply comparing which of A and B is better, or a tie. Our annotators achieved an agreement of 0.72 for this pairwise comparison.

Although these numbers show reasonable agreement, we also note that there is a large extent of subjectivity in human evaluation. This noise level also indicates that some prior work [8, 55] that uses a small number of examples for human evaluation might not be reliable enough. We suggest that the community needs to further improve the reliability and scalability of human evaluation for instruction-following models.

## H    Further Investigation of Figure 2

To further investigate the degree to which the number of unique tokens is being used by GPT-4 as a marker of quality, we created a dummy evaluator that compares two model outputs, and assigns a win to the output with more unique tokens. We plot the win-rate calculated using this dummy evaluator against the win-rate calculated using GPT-4 in Figure 6.

We find that while the dummy evaluator generally over-estimates the win-rate, the trend is still strikingly linear. We note that the $R^2$ for the trendline is .91, suggesting that the unique token count explains a large proportion of the variance in the win rates. Based on this, we believe that the number of unique tokens is certainly a key preference that GPT-4 cares about in its evaluation, although it is still not the only important feature.

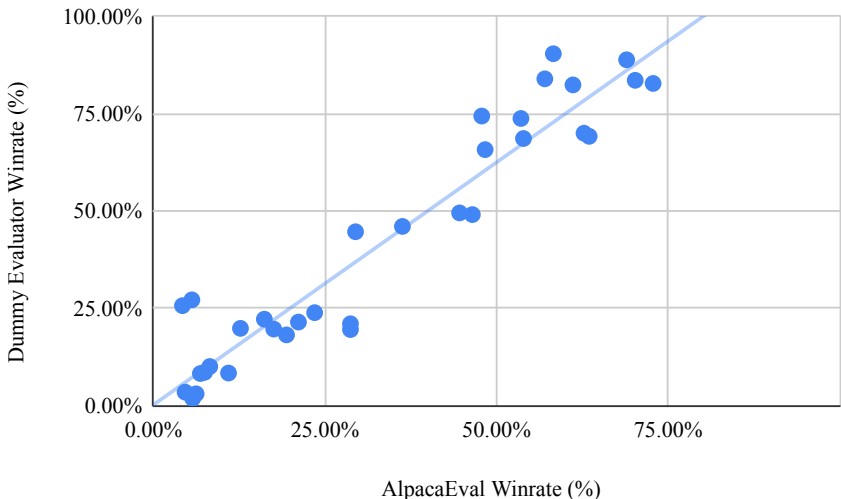

Figure 6: Win-rate scores of all models judged by the dummy evaluator against win-rate of all models using the GPT-4 evaluator.

## I    Model Licenses

We provide brief information about the licenses of the underlying models we make use of in this work below.

- **LLAMA**: The LLAMA model weights are released under a custom license that allows using the model for non-commercial research purposes.
- **LLAMA-2**: The LLAMA-2 model weights are released under a custom license that allows for commercial and research uses with some limitations (e.g., having less than 700 million monthly active users if used in a commercial application), and explicitly allows for redistribution of the weights.

- **Pythia**: The Pythia weights are released under the Apache-2.0 license.
- **OPT**: The OPT model weights are released under a custom license that allow only using the model for non-commercial research purposes.

