# OpenReview forum: "How Far Can Camels Go? Exploring the State of Instruction Tuning on Open Resources"
_NeurIPS.cc/2023/Track/Datasets_and_Benchmarks — NeurIPS 2023 Datasets and Benchmarks Spotlight_

### Official Review · Reviewer_LRM9 · 2023-06-24

**Rating:** 8
**Confidence:** 4

**Strengths:**

1. The paper has a clear problem formulation that is timely and important. While instruction-tuning is common practice by now, the effect of instruction datasets have not been thoroughly analyzed from a comprehensive set of facets in prior work.

2. The paper offers many important insights. To name a few, the findings around
	- Instruction-tuning can benefit less, or sometimes hurt larger models (L232-234)
	- Many open-source datasets can hurt multilingual performance (Table 3).
	- Highlighting that more effort should be spent in characterizing the diversity/comprehensiveness of instruction datasets.

are important to several stakeholders, from users to developers. While some of these are pronounced in earlier works, I believe the paper does a great job of providing a systematic analysis of all these facets.

3. The evaluation is comprehensive and provides a good point of reference to the community. Both the model evaluations and instruction datasets that are experimented with are broad and well-represents prior work.

4. The provided model checkpoints are an extremely valuable resource for the community. It removes the barrier of entry to conduct further research around the effect of instruction-tuning datasets.


**Additional Feedback:**

Thank you for your great contribution. I encourage the authors, and think that it would benefit the open-source community if authors keep maintaining the evaluation suite and improve it over time.

**Clarity:**

The paper is extremely well written, and the figures and tables clearly convey messages.

**Correctness:**

To me, the claims in the paper appear well-justified with sufficient experimentation.

**Documentation:**

The repository and datasets/models that are released are very well-documented.

**Ethics:**

I do not foresee any ethical issues beyond the discussions in the paper.

**Limitations:**

The paper has Limitation (Appendix A) and Broader Impacts (Appendix B) sections with sufficient discussion. I believe it would be good to acknowledge the lack of evaluations around societal impact, and it is easy to add this point.

**Opportunities For Improvement:**

The paper is already a good contribution, I should say that my questions/suggestions are mostly around things that would be interesting to have.

1. One aspect that is not evaluated is a multi-turn dialogue (this is already acknowledged by the authors in Appendix A), which serves many real-world use cases of LMs. The effect of instruction tuning on multi-turn responses is still an important open question. It would certainly be an improvement to have such an evaluation piece.

2. In my opinion, the paper is missing societal impact evaluations, i.e. some aspects that are not evaluated are Bias/Harms/Toxicity. Could one have used datasets such as TruthfulQA or a Bias evaluation dataset to explore this facet? One could ask whether the human-interaction datasets (e.g. ShareGPT) amplify or reduce toxicity or bias. In my opinion having this would be an improvement, however it could be good to highlight this limitation otherwise.

3. In model-based evaluations, models are compared to Davinci-003, but one could ask whether this makes the evaluation error-prone (e.g. Davinci-003 may have certain shortcomings that some models don’t have, and this setting may over pronounce it). It could improve the results to compare to other models in the same setting (I do understand how this may be costly, though). Otherwise could be good to highlight this (apologies if you already did and I missed it).

4. Figure 2 is specifically interesting, and I think this is somewhat reported in AlpacaFarm. I appreciate the correlation analysis, but I would be curious to see if the claim in L249-251 could be made stronger. My understanding is that right now the correlation is measured between the win rate of a model and the avg. unique tokens. But how about, i.e. How would a dummy model evaluator that merely counts the number of unique tokens generated perform, compared to GPT-4? Is there a way to control this effect? What fraction of the variance does this explain?

5.  The paper may benefit from a discussion around [1], as I believe there are shared messages (e.g. [1] suggests models imitate style properties, and this paper reports unique token length significantly correlating with GPT-4 evals). Although not necessary, it would be interesting to see variants of Figure 2 with some of the style properties investigated in [1].

[1] Gudibande, Arnav, et al. "The false promise of imitating proprietary llms." arXiv preprint arXiv:2305.15717 (2023).


**Relation To Prior Work:**

The paper does a good job of synthesizing the findings in the earlier works and further improving the findings around instruction-tuning. The instruction datasets from earlier works are broad and well-represented. I only mentioned one related evaluation paper [1] above, which is a very recent one that should not be required to add.

[1] Gudibande, Arnav, et al. "The false promise of imitating proprietary llms." arXiv preprint arXiv:2305.15717 (2023).


**Summary And Contributions:**

Open-source models are advancing fast, and instruction tuning plays an important part in improving the model performance and human experience. There is an increasing number of instruction-tuning datasets, and it is unclear what are the tradeoffs between datasets and performance in different downstream tasks. This work systematically analyzes open-source instruction tuning efforts and provides many interesting insights that are of importance to researchers, developers, and users. The paper further releases the model and data suite, which allows researchers to conduct further analyses of the topic.

---

> ### Author Response · Authors · 2023-08-21
>
> Thank you for the positive review and for noting that our work provides important insights and is a good point of reference for the community.
>
> **Bias/Harms/Toxicity Evaluation**
>
> We agree that evaluating models on these aspects is important, and have added additional experiments targeting this - please see part 2 of our general response and our paper updates.
>
> **Comparing to models other than davinci-003 in model-based evaluation**
>
> We agree that further model comparisons would be useful, but are limited by the cost and time taken to process model outputs in the GPT-4 API. To keep costs down and scores comparable across models, we opted to use the setting from AlpacaFarm and compare just to text-davinci-003.
>
> **Stronger investigation of the correlation result in Figure 2**
>
> We agree the result shown in Figure 2 is surprising. We followed your suggestion and wrote a dummy evaluator that compared the number of unique tokens in each response and marked whichever model with the most unique tokens as the winner. Comparing the results from this dummy evaluator with the GPT-4 evaluator, we find that the % win-rate for each model between each evaluator has a correlation of .95. We plot the dummy win-rate against the GPT-4-judged win rate for each model here (Figure 6 in the updated paper), and find that while the dummy evaluator generally over-estimates the win-rate, the trend is still strikingly linear. We note that the R-squared for the trendline is .91, suggesting that the unique token count explains a large proportion of the variance in the win rates. Based on this, we believe that the number of unique tokens is certainly an aspect that GPT-4 cares about in its evaluation, although it still does not entirely explain the AlpacaFarm win rates. We have added this discussion and results in Appendix I of our updated paper.
>
> **Discussion of "The false promise of imitating proprietary LLMs"**
>
> We agree that our work shares some findings with Gudibande et al., especially in finding that certain stylistic qualities can greatly influence model-based evaluations, even while large gaps exist in other benchmark scores. We also find that base models are extremely important to performance. However, unlike Gudibande et al., we argue that there are indeed significant differences between instruction datasets, and that imitation-style datasets close the gap most substantially across evaluation settings, suggesting that further improvements to these datasets could yield models even closer to GPT models. We have added some of this discussion to our related work (Section 6).

---

> > ### Comment · Reviewer_LRM9 · 2023-08-22
> > **Response to the Rebuttal**
> >
> > I thank the authors for their rebuttal and additional experiments. In particular, I appreciate the stronger investigation of the correlation result and new societal impacts evaluations. I also saw the discussion you added to Appendix A, L583-L588 for model-based evaluation and potential limitations. These results do answer my questions / comments.
> >
> > This is another minor comment, but it could benefit the paper to refer to Appendix A (if I'm not missing it) in relevant parts of the text to make it discoverable (e.g. maybe can add a reference to somewhere around L180 for the Davinci-003 & model-based eval limitation).

---

> > > ### Author Response · Authors · 2023-08-24
> > >
> > > Thanks for the additional comment! We will reference that appendix in the main text as you suggested.

---

### Official Review · Reviewer_W1Ez · 2023-07-21
**A good-quality work but not a dataset or benchmark paper**

**Rating:** 6
**Confidence:** 5
**Correctness:** Yes
**Clarity:** Yes

**Strengths:**

The work is a decent exploration of fine-tuning the LLMs, the experiment setup and results are well presented, and the code/model release is complete.

**Additional Feedback:**

please see opportunity for improvements and limitations

**Documentation:**

It's fundamentally not a dataset or benchmark paper. However, the open-sourced code and models seem well documented.

**Limitations:**

1. As mentioned above, the scope of work is ambiguous. **Instruction** usually refers to some text input regarding the procedures or requirements.

2. Though the experiment are complete and the conclusions are achieved, there are hardly any inspiration to the community. The 5 key findings listed in L52-L62 are already widely agreed.

**Opportunities For Improvement:**

1. First of all, I don't think the work should be submitted to the dataset and benchmark track. Could authors explain why the work is suitable for this track?

2. The title is kind of confusing. I understand it says **how far can camels go** because of its' finetuned model has a name related to camels, but this sentence in the title is still misleading.

3. The scope of the work, **instruction tuning** is ambiguous. From the explanation in Sec.2.1, it looks like instruction tuning is nothing but finetuning. Then the question raises that why those specific dataset are selected, since the finetuning is general enough to apply to almost every supervised learning dataset.

**Relation To Prior Work:**

No. Despite the surge of LLM-related works, the authors did not discussed about existing studies on fine-tuning large models, and why this work is novel (aside from using a different set of dataset)

**Summary And Contributions:**

The work finetuned the LLAMA family models on multiple instruction dataset in the supervised manner.

The main contribution of the work is the open-sourced code and finetuned model checkpoints.

---

> ### Author Response · Authors · 2023-08-21
>
> Thank you for your review and for noting that our work is well-presented.
>
> **Why is this work suitable for this track?**
>
> We thank the reviewer for raising this question. We refer the reviewer to part 1 of our general response on why this study fits into the [scope of this track](https://neurips.cc/Conferences/2023/CallForDatasetsBenchmarks) for “data-centric machine learning research”, “audits of existing datasets” and “carefully and thoughtfully designed (collections of) datasets based on previously available data”.
>
> **Confusing title**
>
> We apologize for the confusion! The title is somewhat of a play on the fact that, at the time of writing, many released models were using names from the Camelidae biological family, likely due to the fact that they built off the LLaMa models (e.g., alpaca, dromedary, vicuna, guanaco). As we draw from some of the datasets associated with these models (alpaca, vicuna, guanaco) in our work, we are assessing “how far these camelids can go (in evaluation)” by scaling the model size and combining the best resources.
>
> **Instruction tuning is ambiguous**
>
> We thank the reviewer for noting that our definition is perhaps too broad. We have adjusted section 2 introducing instruction tuning to specify that it focuses more on requests that include some indication of the task to be performed within the request itself, which is different from typical task-specific supervised finetuning. We note that we are building off the definitions used in prior, widely-adopted works such as Flan [1], Natural Instructions [2], and InstructGPT [3]. Like the reviewer pointed out, existing datasets can be converted into instruction-tuning format and then be used to finetune models in a supervised manner. While it looks like “nothing but finetuning” in terms of the format, there are many critical problems that can impact the finetuned model’s generalizability on the training tasks and completely unseen tasks, which we hope our work can provide some insights for.
>
> **The key findings are already widely agreed**
>
> We respectfully disagree with the reviewer that our findings are not inspirational for the community. First, we note that there has been a substantial amount of time between submission and current time (1.5 months, which is long for the current LLM community), and at the time our work was being developed, there was significant debate about whether using GPT-distilled datasets was useful [4], how many instruction datapoints could result in good quality models [5], and how to properly evaluate these models [6]. We also argue that even if some of our findings feel obvious in hindsight, clearly and carefully benchmarking and showing these findings in a reproducible, comprehensive manner is still important and valuable for the community.
>
> In addition, we note that reviewers 9TsL and LRM9 both suggest that our work is timely and useful for the community. This shows that other researchers in the field believe that this work does provide utility to the community.
>
>
> [1] Finetuned Language Models Are Zero-Shot Learners (Wei et al., ICLR 2022)
>
> [2] Cross-Task Generalization via Natural Language Crowdsourcing Instructions (Mishra et al., ACL 2022)
>
> [3] Training language models to follow instructions with human feedback (Ouyang et al., NeurIPS 2022)
>
> [4] The False Promise of Imitating Proprietary LLMs (Gudibande et al., Arxiv 2023)
>
> [5] LIMA: Less Is More for Alignment (Zhou et al., Arxiv 2023)
>
> [6] Large Language Models are not Fair Evaluators (Wang et al., Arxiv 2023)

---

> ### Comment · Reviewer_W1Ez · 2023-08-29
> **Thanks for the response**
>
> Based on the revision and authors response, I raised the score.
>
>
> Regarding the scope, despite no other reviewers has the concern, I still think it's a more model centric than a dataset oriented study.

---

### Official Review · Reviewer_9TsL · 2023-07-24
**A timely benchmark paper with useful insights, though it's a bit thin on the why part**

**Rating:** 8
**Confidence:** 4
**Correctness:** Yes
**Clarity:** It's pretty easy to follow

**Strengths:**

- Understanding what matters in instruction tuning of LLMs is highly important and timely
- Comprehensive evaluation
- Useful insights
- New instruction-tuned model that may be helpful for future research

**Additional Feedback:**

Minor:
- some of the references are missing the venue (mostly arxiv preprints)
- I'm not sure if it has been made clear in the paper why the vanilla LLama was not evaluated for open-ended instruction following. Is it because it's not "chatty" enough w/o instruction tuning? Better to make it more clear if not already.

**Documentation:**

Yes, the codebase is already released with reasonable documentation

**Limitations:**

The provided limitations section covers it well.

**Opportunities For Improvement:**

- More in-depth exploration and discussion on reason behind the main observations

**Relation To Prior Work:**

Yes, the positioning in the literature is accurate

**Summary And Contributions:**

This paper compiles most of the public instruction tuning datasets and conducts a comprehensive study to investigate the impact of the base LLM and the nature and source of the instruction tuning data. Both benchmark-based evaluation (for testing various core capabilities) and open-ended instruction following evaluation are conducted, which provide a more complete picture. A new instruction-tuned LLM, by tuning LLaMa 65B on 7 of the instruction datasets, is also released, which is shown to work better than existing ones. Several interesting observations are drawn from the investigation (not repeating them here), which I believe provides timely, much needed information for the community. The main weakness of the current paper is on the **why** part, which is barely touched upon. For example, why instruction tuning often hurts performance on some datasets? Why SuperNI (on GSM and BBH particularly) and Self-instruct seem to be trailing behind in particular? Having more in-depth exploration and discussion on such issues would be tremendously helpful, but I also understand that some of these are big open questions in themselves. Overall, this still presents a good benchmark paper.

---

> ### Author Response · Authors · 2023-08-21
>
> Thank you for the positive review and for noting that our work is timely and important for the community! We have fixed the references with missing venues in our updated paper. We address the questions here:
>
> **Why do some datasets hurt and how can we explain their performance?**
>
> We agree that it is important to discuss and investigate why certain datasets help and hinder model performance, and we point the reviewer to part 2 of our response to review #3, where we discuss some reasons we believe certain datasets hurt performance on certain benchmarks. We have updated our paper with this discussion (Section 5.1). We believe a more in-depth and algorithmic exploration of ‘instruction data diversity and quality’ is a large open area for future research, and our own work provides a useful collection of resources and reproducible results for further investigating this.
>
> **Why not evaluate vanilla LLaMa on AlpacaFarm?**
>
> We did not choose to do this as the vanilla model is not trained to respond to user queries and tends to overgenerate. While few-shot prompting could fix this, initial experiments showed very poor results with the vanilla models and hence we didn't include them. We have added a clarification around this in our updated paper.

---

### Official Review · Reviewer_rWkW · 2023-07-24
**Great evaluation effort for open-source instruction tuning datasets**

**Rating:** 7
**Confidence:** 1
**Clarity:** The paper is well written.

**Strengths:**

1. The authors perform a comprehensive study on open-source instruction-tuning datasets and demonstrate the impact of instruction-tuning with these datasets on open models of various sizes. It shows that instruction tuning is in general beneficial to improve model performance, especially for smaller models.

2. This work suggests a benchmark-based evaluation method for core reasoning and fact-recall skills in addition to the model or human-based evaluation methods which can be subjective. It also shows how model-based evaluation might be biased towards responses with more unique tokens. The results on human-based evaluation show inconsistent evaluation on two different metrics (acceptance rate and pairwise preference rates), which are interesting and might suggest some inconsistency in human evaluation.

3. The TULU model checkpoints, training and evaluation scripts are released on Github.


**Additional Feedback:**

No additional feedback.

**Correctness:**

The evaluated datasets and the evaluation methods and benchmarks are collected from prior work. They seem to be correct.

**Documentation:**

Yes, a url is provided and the code is released under an apache-2.0 license.  The model cannot be used for commercial purpose and no maintenance plan is povided.


**Limitations:**

1. This work didn't introduce new instruction tuning datasets or new benchmarks for evaluating the ability of LMs. It can be useful to

**Opportunities For Improvement:**

1. Might be interesting to extend the instruction-tuning evaluation to the newly released llama2-based models and see how good it is compared to proprietary models.
2. It will be helpful to perform an in-depth analysis of why some instruction tuning datasets hurt the task performance and what its implication is on the application of the finetuned models. Instead of fine-tuning one model with all datasets, would creating an ensemble of domain-specific models or dynamically loading the weights based on prompt inputs lead to better performance in open models?

**Relation To Prior Work:**

Yes, it provides a more comprehensive evaluation on various instruction tuning datasets using a reasonable metrics and evaluation methods.

**Summary And Contributions:**

This work evaluates the performance of instruction-tuned language models of various sizes on a range of open-source datasets. It systematically and comphrensively evaluates the capability of the models on factual knowledge, reasoning, multilinguality, coding, and open-ended instruction following using a collection of human, model, and benchmark-based metrics. For model-based evaluation, this work extends the output length from 300 to 2048 to avoid cut-off in generation.
It then introduced their best performing models TULU which are LLaMa models trained on Human+GPT data mixture that demonstrate better performance over LLaMa on the tasks evaluated. The key findings in the evaluation indicate that domain-specific instruction dataset is effective at improving the model performance in the corresponding domain.  However, there is no single dataset that offers optimal performance across all evaluations. On the other hand,  to achieve the best overall performance on benchmark tasks, the combination of datasets still proves to be more effective. Furthermore, the research highlights the significant influence of the base model. Better base models, whether they are trained on more tokens or are larger in size, consistently outperform smaller ones in various evaluations.

---

> ### Author Response · Authors · 2023-08-21
>
> Thank you for your positive review and for noting that our paper is well-written and comprehensive.
>
> **LLaMa-2 results**
>
> We have added results using LLaMa-2 as a base model into the updated paper - please see part 2 of our general response for details.
>
> **Why do some datasets hurt?**
>
> We agree that it is interesting that some datasets degrade model performance rather than improve it. We believe that this happens when:
> - The dataset/mixture and the evaluation task diverge substantially. Most datasets hurt multilingual performance likely due to their English focus, and datasets without chain-of-thought data (e.g., SuperNI, Self-Instruct) do poorly on reasoning tasks that rely on chain-of-thought prompting for evaluation.
> - The data quality is poor. Self-Instruct and unnatural instructions were (relatively) early attempts at instruction data generation, and made use of weaker models than e.g. alpaca to generate their data. As such, it follows that they may include data that is detrimental for performance.
> - Output style differs greatly. Some evaluations, such as Codex-Eval and AlpacaFarm, rely on long generations from models, which stands in contrast to the largely fairly short responses found in SuperNI and Flan, which tend to be at most a handful of tokens.
> We note that the investigation of how to further improve instruction datasets is an ongoing area of research, and fully explaining the results we show is beyond the scope of this work. We have added some further discussion on why we believe some datasets are better than others into our final version (sec. 5.1).
>
> **Examining ensembling / other methods**
>
> While we agree examining ensembling methods is interesting in the realm of instruction tuning, we believe this is out of the scope of our work, which aims to benchmark and analyze existing resources rather than introduce entirely new methods on its own. There are some recent works studying this direction (e.g., MoE instruction tuning [1] or LoraHub [2]). We hope that our results and codebase serve as solid foundations for future work exploring this.
>
> **Didn’t introduce new datasets or benchmarks**
>
> Given that the LLM community has rolled out many instruction datasets recently but many of them didn’t compare with each other systematically, we believe that a systematic empirical study of these existing resources is at least as important as introducing new ones. We also discussed in the general response why our study fits into the [scope of this track](https://neurips.cc/Conferences/2023/CallForDatasetsBenchmarks) for “data-centric machine learning research”, “audits of existing datasets” and “carefully and thoughtfully designed (collections of) datasets based on previously available data” - please see part 1 of our general response.
>
> **Maintenance plan**
>
> We agree that having a plan for maintaining the codebase is important. We plan to keep updating the codebase with new datasets, models, and techniques, as they come out, and are actively making use of the codebase in follow-up projects and experiments. We have added this plan to our codebase documentation.
>
> [1] Mixture-of-Experts Meets Instruction Tuning: A Winning Combination for Large Language Models (Shen et al., arXiv 2023)
>
> [2] LoraHub: Efficient Cross-Task Generalization via Dynamic LoRA Composition (Huang et al., arXiv 2023)

---

### Official Review · Reviewer_hHUW · 2023-07-29
**How Far Can Camels Go? Exploring the State of Instruction Tuning on Open Resources**

**Rating:** 8
**Confidence:** 4

**Strengths:**

* In my personal opinion, the current submission is of the camera-ready quality; it is very easy to read.
* The experimental design is diverse: the authors consider multiple types of (i) open-source instruction-tuning datasets, (ii) downstream tasks for evaluating the models, and (iii) metrics for evaluating the models.
* The instruction-tuned models are publicly released.

**Additional Feedback:**

* Line 187: add reference to the appendix (Figure 5)
* Any plans on performing similar experiments on LLaMa-2? (mentioning this in the future work?)

**Clarity:**

* The paper is very well written and organised.

**Correctness:**

As mentioned above, the experiments are described in detail, and the reproducibility is supported by an open-source code and models on GitHub.

**Documentation:**

The authors provide sufficient information about SFT, datasets, and evaluation setup. More details and evaluation results are provided in the appendices.

**Limitations:**

The authors comprehensively describe the limitations and broader impact in the appendices. These sections could also mention the subjectivity of individual annotators/difficulty of objectively evaluating generated outputs (related to the experiments in Section 4.3, 5.4).


**Opportunities For Improvement:**

* While it is understandable that the evaluation setup can always be extended, I would suggest considering evaluation of the models on social bias/risks datasets in the revision. It can also be mentioned in the "Broader Impact" section that developers/practitioners should test the models on safety/risks related to their end deployment scenario. As different versions of the models are released, the authors could also mention that no additional SFT costs on the specific datasets are required for the community.
* The author could provide information about the model licenses in the paper.


**Relation To Prior Work:**

Section 6, given the space constraints, sufficiently describes how the work relates to previous contributions, and which/how the related approaches are adopted in the work.

**Summary And Contributions:**

This paper is devoted to analysing the impact of supervised finetuning (SFT) of multiple LLaMa-1 models on single and mixture of open-source instruction-tuning datasets in English. The results of evaluating the models on five groups of tasks (factual knowledge, reasoning, multilinguality, code generation, and open-ended instruction following) show that some datasets help the models to acquire specific skills, but no single dataset/mixture of datasets provide the best performance on the considered tasks.

The contributions of the paper are the folllowing:
- conduction of the SFT & evaluation experiments, and their comprehensive analysis
- releasing four versions of the resulting models (Tuelu): 7B, 13B, 30B, 65B

---

> ### Author Response · Authors · 2023-08-21
>
> Thank you for your positive review and for noting that our current submission is camera-ready quality! We have fixed the reference in Figure 5 and added the information on model licenses into a new appendix section (Appendix H).
>
> **Evaluation of social bias/risks**
>
> We agree that evaluating models on societal biases/risks is a very important aspect and have added experiments for our models on Toxigen and TruthfulQA, two popular datasets for testing a model’s likelihood to generate toxic language and misinformation respectively. Please see part 2 of our general response for details! The results have been added to our updated paper in Section 5.3 and Appendices J & K.
>
> **LLaMa-2 results**
>
> We have added results using LLaMa-2 as a base model instead into the updated paper - please see part 2 of our general response for details. We note that LLaMa-2 was not released at the time of submission, so we were unable to use it for our original submission.

---

> > ### Comment · Reviewer_hHUW · 2023-08-29
> > **Response on the revision**
> >
> > I understand that LLaMa-2 was not released at the time of submission, so I suggested considering experiments on LLaMa-2 for the future work. The authors, however, did a great work on adding the LLaMa-2 in the revision. I confirm that the revision has addressed all my concerns/suggestions, and I see no reason not to accept this paper. I also have read other reviewers' comments and the authors' replies. My score remains the same. Thank you.

---

### Author Response · Authors · 2023-08-21
**General response to reviewers**

We thank all the reviewers for their comments, with multiple reviewers agreeing that our paper is clearly-written, contains useful and timely results for the community, and provides a systematic and comprehensive analysis of instruction-tuning resources.

**Question about “why is this work suitable for this track?”**

While our work does not introduce new datasets, we note that the track's [call for papers](https://neurips.cc/Conferences/2023/CallForDatasetsBenchmarks) explicitly says it “welcomes all work on data-centric machine learning research (DMLR)” (our work being definitely data-centric), and it also lists "audits of existing datasets", "in-depth analyses of machine learning challenges", and "carefully and thoughtfully designed (collections of) datasets based on previously available data" under the scope of this track, which precisely describe our focus. Additionally, the [original blog post introducing the track in 2021](https://neuripsconf.medium.com/announcing-the-neurips-2021-datasets-and-benchmarks-track-644e27c1e66c) states they wish for "systemic benchmarking of algorithms across a wide range of datasets".

We believe that given the extreme pace of the LLM community, performing a comprehensive benchmarking of existing datasets (and analyzing the results), and releasing the code, models, and results, is useful and important work for the community (and we note that reviewers 9TsL and LRM9 both say that our work is timely and useful for the community).

**Paper updates and new results**

We have made a number of improvements to our paper based on reviewer comments, and have run extra experiments around (1) using the newly-released LLaMa-2, and (2) evaluating models for toxicity and truthfulness.

In the updated paper, we mark major updates made in response to reviews in blue. Here we highlight the key findings.

- **LLaMa-2 Results**

Since submitting our work, LLama-2 was released, which provides a theoretically much stronger base model for instruction tuning, being pretrained on a larger number of tokens than LLaMa 1 (among other things). We validate this empirically by training LLama-2 7b and 13b on our Human+GPT (aka ‘Tulu’) mix, and find that the resulting models are significantly stronger than their LLaMa-1 counterparts (please see Table 5 in our updated paper for detailed breakdown):

| Model  | Avg Perf. on the 6 benchmarks  |
|:--------|:------------------------------------:|
|Tulu (LLaMa-1) 7B          |     38.8                   |
|Tulu-2 (LLaMa-2) 7B       |     46.5                    |
|Tulu (LLaMa-1) 13B         |      44.5                  |
|Tulu-2 (LLaMa-2) 13B     |      52.9                  |

This further strengthens our finding that base model quality is extremely important for models' performance after instruction tuning. We plan to additionally train the 70b size LLaMa-2, but cannot finish this within the rebuttal time period due to compute limitations.

We will release our trained LLaMa-2 models shortly and have already updated our codebase to support them.

- **Evaluation of societal biases/risks**

We agree with reviewers that evaluating models for their ethical and other risks is important, and have performed two additional evaluations targeting this: ToxiGen [1] and TruthfulQA [2]. We added the results to our paper (Section 5.3, and Appendices J & K).

For Toxigen, we sample 500 hateful prompts for each group in the dataset, and use the provided roberta-based classifier to determine if completions are toxic or not. We find that all instruction-tuning datasets reduce toxicity to some degree, with ShareGPT and GPT4-Alpaca performing the best. This is likely due to the inclusion of examples of refusals in the distilled data. Additionally, we find that our Tulu mix keeps the lowered toxicity, with Tulu-13B actually achieving a lower toxicity score than all other same-sized models.

For TruthfulQA, we follow the LLaMa-2 paper setup [3] and report the rate of model outputs being truthful, informative, and both (primary metric). We find that instruction tuning generally improves the truthfulness of the pretrained language models across datasets and models. However, the improvement varies a lot across datasets. Interestingly, models can easily learn to be truthful from the distillation data from GPT4 (GPT4-alpaca and ShareGPT), while noisy data (Self-Instruct) can hurt the truthfulness. As we scale up the model size, while larger models are generally more truthful, it seems they tend to hedge more (providing less information as measured by %info), so the rate of being both informative and truthful doesn't see significant improvement from using larger models.

[1] ToxiGen: A Large-Scale Machine-Generated Dataset for Adversarial and Implicit Hate Speech Detection (Hartvigsen et al., ACL 2022)

[2] TruthfulQA: Measuring How Models Mimic Human Falsehoods (Lin et al., ACL 2022)

[3] LLaMa 2: Open Foundation and Fine-Tuned Chat Models (Touvron et al., Arxiv 2023)

---

### Decision · Program_Chairs · 2023-09-22

**Decision:**

Accept (Spotlight)

**Comment:**

This paper presents thorough analysis on a wide range of instruction-tuned LMs.  The experiments are timely and the results will provide interesting insights to researchers in this field.